# ZERO-SHOT GENERALIZATION OF GNNS OVER DISTINCT ATTRIBUTE DOMAINS

## ABSTRACT

Inductive GNNs are able to generalize across graphs with the same set of node attributes. However, zero-shot generalization across attributed graphs with disparate node attribute domains remains a fundamental challenge in graph machine learning. Existing methods are unable to effectively make use of node attributes when transferring to unseen attribute domains, frequently performing no better than models that ignore attributes entirely. This limitation stems from the fact that models trained on one set of attributes (e.g., biographical data in social networks) fail to capture relational dependencies that extend to new attributes in unseen test graphs (e.g., TV and movies preferences). Here, we introduce STAGE, a method that learns representations of *statistical dependencies* between attributes rather than the attribute values themselves, which can then be applied to completely unseen test-time attributes, generalizing by identifying analogous dependencies between features in test. STAGE leverages the theoretical link between maximal invariants and measures of statistical dependencies, enabling it to provably generalize to unseen feature domains for a family of domain shifts. Our empirical results show that when STAGE is pretrained on multiple graph datasets with unrelated feature spaces (distinct feature types and dimensions) and evaluated zero-shot on graphs with yet new feature types and dimensions, it achieves a relative improvement in Hits@1 between 40% to 103% for link prediction, and an 10% improvement in node classification against state-of-the-art baselines.

## 1 INTRODUCTION

Zero-shot generalization, or a model's ability to handle unseen test data without additional training or adaptation, has long been a key objective for AI systems (Larochelle et al., 2008; Xian et al., 2017; Wang et al., 2022). An essential prerequisite to zero-shot generalization is for models to learn prediction rules that can be used across entirely different sets of features (or attributes) at test time. This challenge has been addressed in other domains, such as natural language, through processes that can split up any data into a pre-determined set of tokens (Samuel & Øvrelid, 2023).

However, in the context of graph data with node attributes, zero-shot generalization presents unique challenges. First, features can vary significantly between graphs in different domains. Any method must be able to handle heterogeneous node features, including mixtures of continuous and categorical features, as well as features that are highly dataset specific such as *RAM* on an electronics store and clothes *size* in a department store, as shown in Figure 1. Second, the interpretation of certain features (such as *size*) is often context-dependent, requiring graph learning methods that can jointly model both the topology and the node attributes. *These challenges make it hard to define a unified input space allowing graph models to zero-shot generalize to unseen attributed graphs.*

For these reasons, pretraining general purpose graph models remains an open challenge. This is reflected in the fact that generalizing to new attribute domains is rarely tackled in zero-shot settings (as we discuss in Section 5). One strategy in such cases is to ignore node attributes altogether, focusing on learning node and edge relations. Alternatively, some works *textify* node features (or the entire graph) using pre-trained text embedding models (Chen et al., 2024a; Huang et al., 2023; Liu et al., 2024; Zhang et al., 2023). However, this latter approach is limited by the text encoder, which may overlook significant predictive signal in numerical attributes (Collins et al., 2024; Gruver et al., 2024; Schwartz et al., 2024), and the inclusion of text encoders often makes end-to-end training challenging.

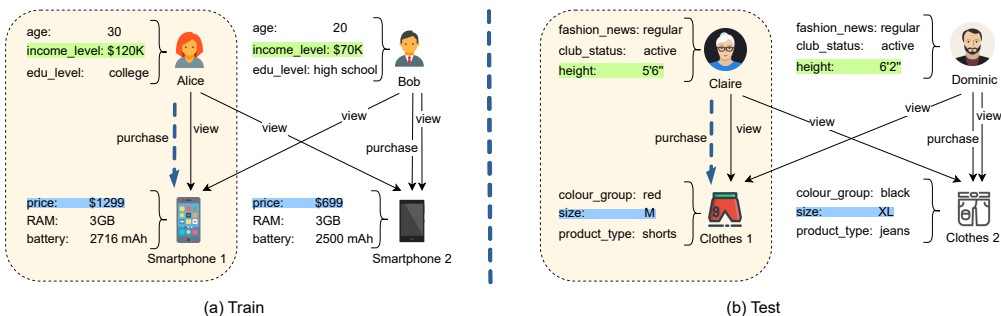

(a) Train                         (b) Test

Figure 1: The task of zero-shot generalization to attributed graphs with unseen attribute/features. This task is challenging because features in the train and test graphs have different semantics. Moreover, real-world graphs typically contain a mixture of continuous and categorical features. Nevertheless, features associated with an edge can be highly correlated (e.g. income level is positively correlated to phone price in (a)). **Our proposed method, STAGE, jointly learns both the *statistical dependencies* among features and the graph structure, and leverages the analogous statistical dependencies between features in the test graph (e.g. the positively correlated height and clothes sizes in (b)) to perform zero-shot transfer of distinct attribute domains.**

Another approach is to focus on test-time adaptation for the GNN node embeddings, accepting that the models representations may degrade on unseen attribute domains. This can be done, for example, by applying linear models to node embeddings using closed-form maximum-likelihood solutions (Zhao et al., 2024). Whilst test-time adaptation may prove to be useful in conjunction with domain generalization, it does not address the fundamental problem of domain generalization itself, which is to bestow the pretrained GNN model with the ability to produce useful representations of any graph.

In this work, we introduce STAGE (**S**tatistical **T**ransfer for **A**ttributed **G**raph **E**mbeddings), a novel design for encoding node features to address the challenges of generalization to new attribute domains on graphs. The core idea behind STAGE is to transform raw node attributes, which exist in an "absolute" natural space that can vary arbitrarily across graphs, into a *relative* space that captures their statistical dependencies over graph edges. For instance, Figure 1 illustrates purchases triggered by positive correlations between the features in both domains. More specifically, STAGE builds a graphical representation of such statistical dependencies that is invariant to three transformations commonly observed when transferring across distinct attribute domains—changes in attribute values, permutations of attribute dimensions, and permutations of node identities—while preserving the maximal amount of information about the raw attribute values. In particular, STAGE transforms node features into edge embeddings through a two-step process:

1. ***STAGE-edge-graph* construction:** For each edge in the original graph, STAGE constructs an *STAGE-edge-graph* (such as illustrated in Figure 2(b), a fully connected weighted graph whose

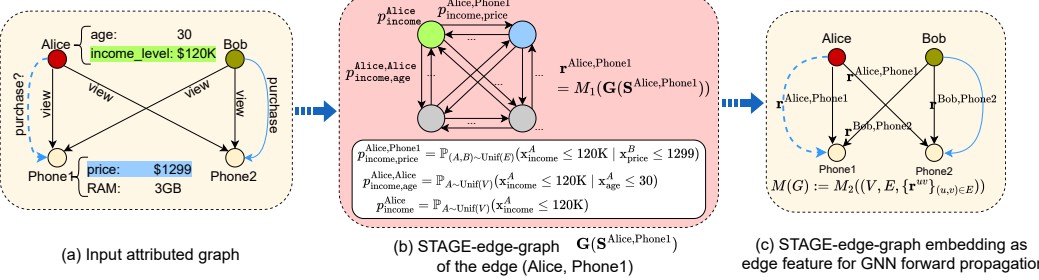

(a) Input attributed graph     (b) STAGE-edge-graph  $\mathbf{G}(\mathbf{S}^{\text{Alice,Phone1}})$     (c) STAGE-edge-graph embedding as
                                  of the edge (Alice, Phone1)                       edge feature for GNN forward propagation

Figure 2: Given an input attributed graph $G$ (a), STAGE builds a *STAGE-edge-graph* (b) for every edge in $G$. Nodes in a STAGE-edge-graph correspond to individual features of the two edge endpoints, and the node and edge attributes are the empirical marginal and conditional probabilities of feature values (Equations (2) and (3)). STAGE then applies the intra-edge GNN on *STAGE-edge-graph* (b) to obtain an edge-level embedding for each input graph edge, and finally applies the inter-edge GNN to obtain the input graph representations leveraging the set of edge-level embeddings (c).

nodes correspond to the features of the two endpoints (if there are $m$ features for each node, the *STAGE-edge-graph* has $2m$ nodes). The edge weights are the conditional probabilities later introduced in Equation (2), and the node attributes are assigned the marginal probabilities later introduced in Equation (3).

2. **Edge embedding addition:** As illustrated in Figure 2(c), for each edge of the original graph, STAGE associates an embedding of a standard GNN architecture applied to that edge's *STAGE-edge-graph*. As GNNs can operate on variable-sized graphs, STAGE is capable of learning over feature spaces of varying dimensions.

STAGE then adds these embeddings to their respective edges as edge features (as illustrated in Figure 2(c)), and removes the original node features before applying a second GNN to process the entire modified graph, now featurized only by the new embeddings as edge features. This allows STAGE to end-to-end pretrain on multiple graph datasets from diverse feature domains for a single task. At test time, it can zero-shot generalize to solve that task on *any* graph, regardless of the novelty, unobserved nature, or varying-size of their node feature spaces.

In order to shift from embedding node features to embedding statistical dependencies on edges, the *STAGE-edge-graph* is constructed using principles established by Bell (1964) and Berk & Bickel (1968), which link the theory of maximal invariants to measuring statistical dependencies between random variables. Namely, we prove that any function that can measure statistical dependencies of node features from totally ordered sets (e.g., $\mathbb{R}^d$, $d \geq 1$), can be described as a graph regression task on a variant of our *STAGE-edge-graph*, as long as its GNN encoder is sufficiently expressive. This means that our method can learn expressive representations independent of the attribute domain for a class of domain shifts, allowing generalization between graph tasks that rely on the interplay between attribute dependencies and graph structure. Note that we *do not* prove generalization between arbitrary graphs, since there are clear worst-case examples for which transfer is impossible.

Our experiments show that STAGE substantially improves zero-shot out-of-domain generalization against state-of-the-art baselines for both link prediction and node classification tasks on multiple datasets. Predicting future customer activity on an e-commerce website (link prediction) involving six distinct stores selling *beds, desktops, refrigerators, smartphones, shoes* and *clothes and apparel* from another (H&M), STAGE achieves up to a 103% improvement in Hits@1 over the best state-of-the-art baselines. Additionally, in node classification, by pretraining on Friendster and zero-shooting into Pokec, STAGE outperforms state-of-the-art by approximately 10%. Notably, STAGE's performance improves as the number of training domains increases, showing it is currently the only method capable of learning generalizable patterns across distinct feature domains during pretraining.

These results underscore STAGE's ability to learn transferable representations that are robust to variations in node attribute domains. By embedding feature dependencies rather than their absolute values, STAGE enables robust zero-shot predictions across entirely new and unseen graph domains.

## 2 STAGE: ZERO-SHOT GENERALIZATION TO DISTINCT ATTRIBUTE DOMAINS

GNNs typically assume that the node feature dimensions of training and test graphs correspond to the same semantics, restricting their applicability to graphs with novel or misaligned feature dimensions at test time. To overcome this limitation, we introduce STAGE, a novel architecture that enables knowledge transfer across graphs with distinct feature spaces.

We are given an attributed graph $G = (V, E, \boldsymbol{X})$ where $V$ is the set of nodes, $E$ the set of edges, and $\boldsymbol{X} = \{\boldsymbol{x}^v\}_{v \in V}$ the set of node features $\boldsymbol{x}^v$ for each node $v \in V$. We assume all $\boldsymbol{x}^v$ belong to some measurable space of dimension $d \geq 1$. To design a model capable of generalizing to test graphs that may have node features living in a different space than $\boldsymbol{X}$, we design a projection map that transforms the node features $(\boldsymbol{x}^u, \boldsymbol{x}^v)$ of an edge $(u, v) \in E$ into a fixed dimensional pairwise embedding

$$\mathcal{P} : (\boldsymbol{x}^u, \boldsymbol{x}^v) \mapsto \boldsymbol{r}^{uv} \in \mathbb{R}^k, \quad k \geq 1. \tag{1}$$

The choice to consider pairwise embeddings allows STAGE to model relations between features belonging to different nodes. For instance, modeling the relation between customer node Alice and product node Smartphone1 (Phone1) in Figure 2(a). We design the mapping $\mathcal{P}$ by building a graph based on the pairwise *pdf* feature descriptors. Viewing node features through their *pdf*s is a crucial step as it transforms potentially non-aligned node feature spaces into a universal space of densities.

Concretely, let A and B be a random pair of nodes jointly and uniformly sampled from the edge set, $(A, B) \sim \text{Unif}(E)$. Let $x_i^A$ denote the random variable of the $i$-th feature value of random node A, and $x_j^B$ the $j$-th feature value of random node B. Given a specific pair of distinct nodes $u, v \in V$ and specific feature values $x_i^u$ and $x_j^v$, we define $p(x_i^u | x_j^v)$ from the conditional probabilities as follows, accounting for mixture of totally ordered (e.g., scalar) and unordered (e.g., categorical) features:

- $p(x_i^u | x_j^v) := \mathbb{P}_{(A,B) \sim \text{Unif}(E)}(x_i^A \leq x_i^u | x_j^B \leq x_j^v)$, if both feature $i$ and $j$ are totally ordered.

For brevity we omit the distribution $(A, B) \sim \text{Unif}(E)$, writing $\mathbb{P}$ instead of $\mathbb{P}_{(A,B) \sim \text{Unif}(E)}$ from now.

- $p(x_i^u | x_j^v) := \mathbb{P}(x_i^A = x_i^u | x_j^B \leq x_j^v)$, if feature $i$ is unordered and feature $j$ is totally ordered.
- $p(x_i^u | x_j^v) := \mathbb{P}(x_i^A \leq x_i^u | x_j^B = x_j^v)$, if feature $i$ is totally ordered and feature $j$ is unordered.
- $p(x_i^u | x_j^v) := \mathbb{P}(x_i^A = x_i^u | x_j^B = x_j^v)$, if both feature $i$ and $j$ are unordered.

If $u = v$, we change the sampling distribution to $(A) \sim \text{Unif}(V)$ and let $B = A$. Everything else remains the same in the definitions above. In practice, all these probabilities can be empirically estimated from the input data. For the node-pair $u, v$ we define a conditional probability matrix $\boldsymbol{S}^{uv}$, with indices $i, j \in \{1, \ldots, 2d\}$, $i \neq j$ as follows:

$$
\boldsymbol{S}_{ij}^{uv} = \begin{cases} p(x_i^u \mid x_j^u) & \text{if } i \leq d \text{ and } j \leq d, \\ p(x_{i-d}^v \mid x_{j-d}^v) & \text{if } d < i \leq 2d \text{ and } d < j \leq 2d, \\ p(x_i^u \mid x_{j-d}^v) & \text{if } i \leq d \text{ and } d < j \leq 2d, \\ p(x_{i-d}^v \mid x_j^u) & \text{if } d < i \leq 2d \text{ and } j \leq d. \end{cases} \tag{2}
$$

and for the diagonal $i = j$ we define,

$$
\boldsymbol{S}_{ij}^{uv} = \begin{cases} p(x_i^u) & \text{if } i \leq d, \\ p(x_i^v) & \text{if } i > d, \end{cases} \tag{3}
$$

where $p(x_i^u) := \mathbb{P}(x_i = x_i^u)$ if $x_i^u$ is unordered and $p(x_i^u) := \mathbb{P}(x_i \leq x_i^u)$ if $x_i^u$ is totally ordered. This diagonal term allows STAGE to also model intra-node feature dependencies.

The matrix $\boldsymbol{S}^{uv}$ is the core node-pair data representation STAGE uses. This matrix is used to define a graph structure which we call a STAGE-edge-graph as illustrated in Figure 2(b), which gives a parwise feature coordinate-level description of the relation between the two endpoint nodes.

**Definition 2.1** (STAGE-edge-graph). Given a pair of nodes $u, v \in V$, a STAGE-edge-graph for $(u, v)$ is a fully connected, weighted, directed graph $\boldsymbol{G}(\boldsymbol{S}^{uv})$ with $2d$ nodes, where node $i$ has a scalar attribute $\boldsymbol{S}_{ii}^{uv}$, and edge $(i, j)$ has a scalar attribute $\boldsymbol{S}_{ij}^{uv}$.

**STAGE algorithm.** As illustrated in Figures 2(b) and 2(c), STAGE uses the STAGE-edge-graph of all edges in a two-stage process to produce attribute-domain-transferable graph representations. First, STAGE uses a GNN to generate expressive embeddings for each STAGE-edge-graph of each edge. After these edge embeddings are added, STAGE removes the original node features. This modified graph is then fed into a second GNN to solve the overall task, producing the final node, link, or graph representation. The two steps of STAGE are as follows.

1. *(Intra-edge)* Each $\boldsymbol{G}(\boldsymbol{S}^{uv})$ is processed independently with a GNN $M_1$ to produce edge-level embeddings $\boldsymbol{r}^{uv} = M_1(\boldsymbol{G}(\boldsymbol{S}^{uv}))$ in Equation (1).

2. *(Inter-edge)* A second GNN $M_2$ processes $G' = (V, E, \{\boldsymbol{r}^{uv}\}_{(u,v) \in E})$, i.e., the original graph equipped with the learned edge embeddings to give a final representation $M(G) := M_2(G')$.

The two GNNs $M_1$ and $M_2$ are trained end-to-end on the task. Note that $M_1$ can be any GNN designed to produce whole-graph embeddings and can take single-dimensional edge features, whilst $M_2$ can be any GNN that can take edge embeddings as input.

**Modelling pairwise relations.** $\boldsymbol{S}^{uv}$ is only computed for *edges* $(u, v)$, and so can only model pairwise relations between nodes connected by an edge. In some cases, such as bipartite graphs, we find it beneficial to add extra edges between nodes of the same type (see Section 4 for details). In general, higher-order relations could also be modelled similarly, albeit at increased complexity. We leave exploration of higher-order relations to future work.

# 3 STATISTICAL UNDERPINNINGS OF STAGE

This section explains how STAGE achieves domain transferability. The central result is to show that STAGE generates representations capable of measuring feature dependencies on graphs. This means that STAGE is able to ignore "absolute" feature values, whilst still generalizing through analogous statistical dependecies of the unseen features in the test graph.

Our first step (Section 3.1) connects measures of statistical dependencies with a novel graph regression task. Then, Section 3.2 shows that our STAGE-edge-graphs (Definition 2.1) can lead to a compact model for this regression, with a variant that is invariant to a class of shifts between train and test feature domains. The following theoretical results are meant to provide insights and are restricted to domains with a fixed number of features to simplify the proofs, extending them to variable size spaces is left as future work. Detailed proofs are provided in Appendix A.

## 3.1 STATISTICAL DEPENDENCE OF NODE-PAIR FEATURES AS A GRAPH REGRESSION

We begin by introducing the framework for building what we call *feature hypergraphs*. We will show that feature hypergraphs can sufficiently encapsulate the statistical dependencies between features, whilst only leveraging the relative orders rather than the numerical values of the feature, enabling it to be invariant to the order-preserving transformations (defined later in Definition A.2) to achieve better domain transferability. Throughout this exposition we assume one feature space defined over a totally ordered set (e.g., $\mathbb{R}^d$ for $d \geq 1$, where the total order $\leq$ is well defined). Since the invariances of unordered sets are a special case (as these do not need order-preserving transformations), this section focuses on totally ordered sets. Before we describe how feature hypergraphs are built, we start with the concept of order statistic, which captures the relative ordering of the feature values.

**Order statistic** (David & Nagaraja, 2004). Let $\mathbf{x}_1, \mathbf{x}_2, \ldots, \mathbf{x}_m$ be a sequence of $m \geq 2$ random variables from some unknown distribution $F$ over a totally ordered set (e.g., a convex set $\mathbb{F} \subseteq \mathbb{R}$). Its *order statistics* are defined as the sorted values $\mathbf{x}_{(1)} \leq \mathbf{x}_{(2)} \leq \cdots \leq \mathbf{x}_{(m)}$, where $\mathbf{x}_{(k)}$ denotes the $k$-th smallest value in the $m$ samples.

Consider a domain with $m$ entities (e.g., products in an appliance store), where each entity is characterized by $d$ features. Specifically, an entity $u$ can be represented by a (row) vector of random feature variables, $\mathbf{x}^u = [\mathbf{x}_1^u, \mathbf{x}_2^u, \ldots, \mathbf{x}_d^u]$. where $\mathbf{x}_i^u$ describes the $i$-th feature of entity $u$ that takes on values from the $i$-th feature space $\mathbb{F}_i \subseteq \mathbb{R}$. With these variables, we define the (random) matrix $\mathbf{X} := [(\mathbf{x}^1)^T, (\mathbf{x}^2)^T, \ldots, (\mathbf{x}^m)^T]^T$ of shape $m \times d$. Alternatively, we can view $\mathbf{X}$ column-wise, where each feature $i$ corresponds to a (column) random vector $\mathbf{x}_i = [\mathbf{x}_i^1, \mathbf{x}_i^2, \ldots, \mathbf{x}_i^m]^T$. Next, we introduce the order statistic for these features: let $\mathbf{x}_{i(k)}$ denote the *$k$-th order statistic* of $\{\mathbf{x}_i^1, \ldots, \mathbf{x}_i^m\}$. For instance, $\mathbf{x}_{i(1)} = \min\{\mathbf{x}_i^1, \ldots, \mathbf{x}_i^m\}$.

Given an input graph $G = (V, E, \mathbf{X})$, we regard it as a sample from some unknown distribution over all attributed graphs with $m$ entities and $d$ features, where $\mathbf{X}$ is a random variable with $\mathbf{X} = [\boldsymbol{x}_1, \ldots, \boldsymbol{x}_d]$. Consider the edges in $E$ as samples of pairs of nodes that give rise to the multiset of edge endpoint features, $\mathcal{E} = \{\{(\boldsymbol{x}^u, \boldsymbol{x}^v) \mid (u, v) \in E\}\}$. Together with the order statistics, we now define the feature hypergraph as follows:

**Definition 3.1** (Feature hypergraph $\mathcal{F}_{\mathcal{E}}$). Given a multiset of edge endpoint features $\mathcal{E} = \{\{(\boldsymbol{x}^u, \boldsymbol{x}^v) \mid (u, v) \in E\}\}$ of $m$ entities with totally ordered feature spaces, the feature hypergraph $\mathcal{F}_{\mathcal{E}}$ is defined as follows. First, we label the graph with $m$. Then,

- For each order statistic $\boldsymbol{x}_{i(k)}$ of feature $i$ and order $k$ ($1 \leq k \leq m$), there are 2 nodes, labeled as $(i, k, 1)$ and $(i, k, 2)$, respectively. In total, there are exactly $2md$ nodes in $\mathcal{F}_{\mathcal{E}}$ (feature values need not be unique). Nodes $(i, k, 1)$ and $(i, k, 2)$ store a single feature to mark their order: $k$.

- Let $o_i(u)$ be the order of the feature value $\boldsymbol{x}_i^u$, i.e., $\boldsymbol{x}_{i(o_i(u))} = \boldsymbol{x}_i^u$. For each pair of edge endpoint features $(\boldsymbol{x}^u, \boldsymbol{x}^v) \in \mathcal{E}$, there is a hyperedge $H_{uv}$ in $\mathcal{F}_{\mathcal{E}}$ defined as

$$H_{uv} := \{(1, o_1(u), 1), (1, o_1(v), 2), (2, o_2(u), 1), (2, o_2(v), 2), \ldots, (d, o_d(u), 1), (d, o_d(v), 2)\}.$$

Our first observation is that the feature hypergraph in Definition 3.1 perfectly captures the order statistics of the set $\mathcal{E}$ but discards the actual values of the features.

We now consider statistical tests that measure dependencies of the endpoint features. As an illustrating example, consider that if $(\boldsymbol{x}^u, \boldsymbol{x}^v) \in \mathcal{E}$ are samples (not necessarily independently sampled) from a bivariate distribution $(\mathbf{x}, \mathbf{x}') \sim F$, one may be interested in testing the hypothesis

$$H_0 : F(\mathbf{x}, \mathbf{x}') = F_1(\mathbf{x}) F_2(\mathbf{x}'),$$

i.e., that $\mathbf{x}$ and $\mathbf{x}'$ are independent. Bell (1964); Berk & Bickel (1968) have shown over totally ordered sets, measures (e.g., $p$-values) of such hypothesis tests, for pairwise independence ($H_0$ above) and more generally higher-order conditional independence between multiple variables, have invariances that simplify the data representation to such a degree that the original values are discarded, retaining only the order relationships between the variable values. And that any such test is therefore a rank test, i.e., it relies solely on indices of the order statistic rather than the numerical values of the features.

Our first theoretical contribution is the observation that *any statistical test that focuses on measuring the (conditional) dependencies of endpoint features in $\mathcal{E}$ can be defined as a graph regression task over the feature hypergraph $\mathcal{F}_{\mathcal{E}}$ of Definition 3.1.*

**Theorem 3.2.** *Given a multiset of edge endpoint features $\mathcal{E}$, the corresponding feature hypergraph $\mathcal{F}_{\mathcal{E}}$ (Definition 3.1) and a most-expressive hypergraph GNN encoder $M_{\theta^*}(\mathcal{F}_{\mathcal{E}})$, then any test $T(\mathcal{E})$ that focuses on measuring the dependence of the endpoint features of $\mathcal{E}$ has an equivalent function $h$ within the space of Multilayer Perceptrons (MLPs) that depends solely on the graph representation $M_{\theta^*}(\mathcal{F}_{\mathcal{E}})$, i.e., $\exists h \in$ MLPs s.t. $T(\mathcal{E}) = h(M_{\theta^*}(\mathcal{F}_{\mathcal{E}}))$.*

Next we show that the hypergraph $\mathcal{F}_{\mathcal{E}}$ can be simplified with STAGE-edge-graph and that it its ability to compute dependency measures can be made invariant to some domain shifts between train and test.

## 3.2 Transferability: STAGE can model measures of statistical dependencies and is invariant to a family of domain transformations

The feature hypergraph $\mathcal{F}_{\mathcal{E}}$ in Definition 3.1 is used to obtain a maximal invariant graph representation via hypergraph GNN. This solution has a high computational cost from the use of hypergraph GNNs. Fortunately, we show that by assigning unique feature identifiers to label the nodes of our STAGE-edge-graphs $\boldsymbol{G}(\boldsymbol{S}^{uv})$ (Definition 2.1), STAGE-edge-graphs are as informative as the corresponding feature hypergraph (Definition 3.1) while preserving the same invariances.

**Theorem 3.3.** *Given the endpoint features $\mathcal{E}$ (Definition 3.1) of a graph $G = (V, E, \boldsymbol{X})$, there exists an optimal parameterization $\theta_g^*, \theta_s^*$ for a most expressive GNN encoder $M^g$ and a most-expressive multiset encoder $M^s$, respectively, such that $M_{\theta_s^*, \theta_g^*}(G) := M_{\theta_s^*}^s\left(\{\{M_{\theta_g^*}^g(\boldsymbol{G}(\boldsymbol{S}^{uv})) : (u,v) \in E\}\}\right)$ such that any test $T(\mathcal{E})$ that measures the dependence of $\mathcal{E}$'s endpoint features has an equivalent function $h$ within the space of Multilayer Perceptrons (MLPs) that depends solely on the graph representation $M_{\theta_s^*, \theta_g^*}(G)$, i.e., $\exists h \in$ MLPs s.t. $T(\mathcal{E}) = h(M_{\theta_s^*, \theta_g^*}(G))$.*

Theorem 3.3 motivates the design of STAGE, which leverages a GNN on STAGE-edge-graphs to obtain edge-level embeddings. However, the use of unique feature identifiers in the STAGE-edge-graphs disrupts invariance to permutations in feature dimensions (e.g., U.S. shoe size appearing as the first dimension in one dataset and U.K. shoe size as the last dimension in another), thereby limiting its domain transferability. More broadly, we now describe all the invariances we want for STAGE to have to be robust to a class of feature domain shifts.

**COGG invariances.** STAGE-edge-graphs facilitates domain transfer to distinct feature domains. Intuitively, the full set of invariances required for domain transferability over $G = (V, E, \boldsymbol{X})$ consists of: (1) invariance or equivariance to transformations of feature values that preserve the order statistic, (2) invariance or equivariance to permutations of feature dimensions (columns of $\boldsymbol{X}$), and (3) invariance or equivariance to permutations of entities (nodes) in the graph, affecting both $V$ (and consequently $E$) and the rows of $\boldsymbol{X}$. These invariances are formally described in Definition A.5 at Appendix A.4 through actions of *component-wise order-preserving groupoid for graphs* (COGG).

We now introduce our final theoretical contribution which establishes that STAGE achieves invariance to COGGs by design. This result shows that STAGE can provably achieve the zero-shot transferability to the class of feature domain shifts defined by COGGs-type transformations.

**Theorem 3.4.** *STAGE (Section 2) is invariant to COGGs (Definition A.5).*

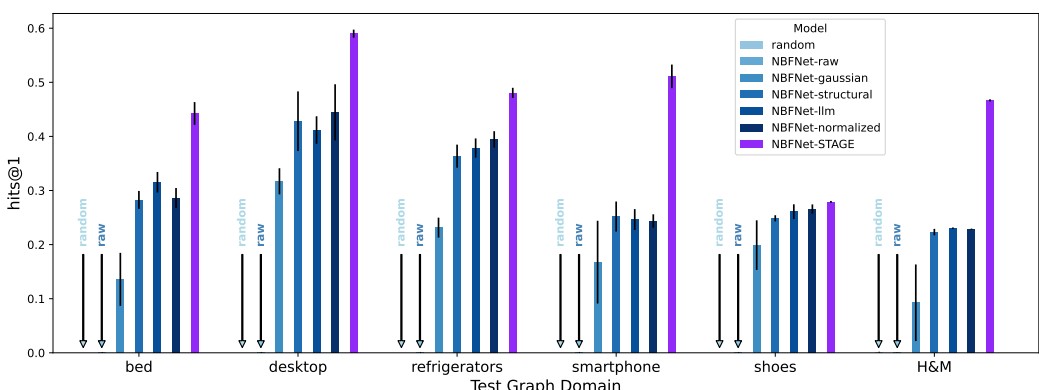

Figure 3: Zero-shot Hits@1 performance (higher is better) of STAGE and baselines, pretrained on four (or five) distinct store domains and evaluated on the held-out domain (or H&M dataset). **NBFNet with STAGE feature encoding consistently achieves the highest zero-shot accuracy across all test domains showing up to 103% improvement**. Error bars show standard deviation over seeds.

The proof sketch is essentially as follows. The first insight is to drop the feature-id labels in STAGE-edge-graphs. This modification sacrifices maximal expressivity (Theorem 3.3), but ensures that STAGE is invariant to permutations of the feature dimensions. Second, STAGE employs a second GNN on the original input graph, using the graph embeddings of the STAGE-edge-graphs as additional edge-level embeddings, while omitting the original node features. This configuration ensures STAGE's invariance to graph isomorphisms. Thus, the entire method is invariant to COGGs.

## 4 EXPERIMENTS

We now show the effectiveness of STAGE across multiple experimental settings. We refer the reader to Appendix C for details and to Appendix F for a complexity analysis and runtime comparison.

**Datasets.** To evaluate STAGE's zero-shot generalization to graphs with unseen attributes, we consider several graph datasets with unique domain-specific node features but a shared task. See Appendix B for more details on these datasets, their tasks, and their construction.

*E-Commerce Stores dataset (link prediction).* We use an E-Commerce dataset from a multi-category store (Kechinov, 2020) containing customer-product interactions (purchases, cart additions, views) over time. We split it into five single product category domains with disjoint customers (*simulating five distinct single-category stores*): *shoes*, *refrigerators*, *desktops*, *smartphones*, and *beds*, each with unique features (e.g., *smartphones* have *display type*, *RAM size*; *shoes* have *ankle height*, *material*) and unique customers. The task is to predict future customer-product interactions from past actions.

*H&M dataset (link prediction).* We use the H&M Personalized Fashion Recommendations dataset (Kaggle, 2021), containing transactions from a large fashion retailer, to evaluate the zero-shot performance of models trained on the E-Commerce Stores dataset. Attributes in H&M dataset except one ("price") are distinct from those in E-Commerce Stores. The task remains predicting future customer-product interactions from past actions.

*Social network datasets (node classification): Friendster and Pokec.* We consider the Friendster (Teixeira et al., 2019) and Pokec (SNAP, 2012) datasets, two online social networks from different regions and user bases. Friendster nodes have features such as *age*, *gender*, *interests*, etc., while Pokec nodes have *public profile status*, *completion percentage*, *region*, *age*, and *gender*. The task is to predict a node feature common to both social networks using network structure and remaining node features. Since only *age* and *gender* are common, we create two tasks: mask and predict *gender*, and mask and regress on *age* (however, age seems to not be predictable as discussed in Appendix D).

**Baselines.** We compare STAGE to several baselines for handling new node features. GNN-RAW: Projects each raw node feature into a fixed-dimensional space via linear transformations, before summing across the projected dimensions. GNN-GAUSSIAN: Use Gaussian noise as node fea-

Table 1: Zero-shot Hits@1 and MRR (higher is better) of STAGE and baselines. **NBFNet-STAGE outperforms all baselines in zero-shot Hits@1 and MRR (including supervised approaches) across the E-Commerce Stores and H&M datasets.** Models were trained on all combinations of four graph domains and tested on the remaining domain for the E-Commerce Stores dataset, and trained on the five E-Commerce Stores domains and tested zero-shot on the unseen H&M dataset. The Structural-Supervised baseline was trained on additional edges in the H&M dataset. Improvement (% **gain**) is calculated as the relative increase in performance of STAGE compared to each baseline.

| Pretraining: E-Commerce Stores | Test: Held-out E-Comm. Store | | | | Test: H&M Dataset | | | |
|---|---|---|---|---|---|---|---|---|
| Model | Hits@1 (↑) | % gain | MRR | % gain | Hits@1 (↑) | % gain | MRR (↑) | % gain |
| random | $0.0026 \pm 0.0000$ | 17615% | - | - | $0.0006 \pm 0.0000$ | 77667% | - | - |
| NBFNet-raw | $0.0000 \pm 0.0000$ | ∞ | $0.0032 \pm 0.0009$ | 15434% | $0.0005 \pm 0.0004$ | 93220% | $0.0059 \pm 0.0011$ | 7871% |
| NBFNet-gaussian | $0.2101 \pm 0.0428$ | 119% | $0.2617 \pm 0.0459$ | 90% | $0.0925 \pm 0.0708$ | 404% | $0.1176 \pm 0.0756$ | 300% |
| NBFNet-structural | $0.3149 \pm 0.0253$ | 46% | $0.3721 \pm 0.0219$ | 34% | $0.2231 \pm 0.0060$ | 109% | $0.2302 \pm 0.0080$ | 104% |
| NBFNet-llm | $0.3226 \pm 0.0190$ | 43% | $0.3830 \pm 0.0145$ | 30% | $0.2302 \pm 0.0015$ | 103% | $0.2365 \pm 0.0021$ | 99% |
| NBFNet-normalized | $0.3269 \pm 0.0213$ | 41% | $0.3844 \pm 0.0159$ | 29% | $0.2286 \pm 0.0010$ | 104% | $0.2341 \pm 0.0018$ | 101% |
| NBFNet-structural-supervised | N/A | N/A | N/A | N/A | $0.1546 \pm 0.0084$ | 202% | $0.2103 \pm 0.0164$ | 124% |
| **NBFNet-STAGE (Ours)** | **0.4606** $\pm 0.0123$ | 0% | **0.4971** $\pm 0.0073$ | 0% | **0.4666** $\pm 0.0020$ | 0% | **0.4703** $\pm 0.0029$ | 0% |

tures (Sato et al., 2021; Abboud et al., 2021). STRUCTURAL: Disregards node features entirely, using only the graph structure. GNN-LLM: Converts node features into textual descriptions and obtains embeddings using a pre-trained encoder-only language model, akin to PRODIGY (Huang et al., 2023). GNN-NORMALIZED: Retains only continuous features and standardize them. For a fair comparison, we use the same underlying GNN architecture for all methods: NBFNet (Zhu et al., 2021c) for the E-Commerce and H&M datasets (link prediction), and GINE (Hu et al., 2020) for social network datasets (node classification). In addition to these baselines, we also evaluate our approach against GraphAny (Zhao et al., 2024), a recent method specifically tailored for domain transferability in node classification tasks, but not applicable to link prediction. In Appendix E, we perform additional ablation studies with alternative GNNs. All models share hyperparameters where applicable, and results are averaged over three seeds. For graphs with node types lacking features (e.g., customers in E-Commerce Stores), we build edges between featured nodes of the same type (e.g., products) based on common connections, forming *STAGE-edge-graph* for these new edges. These edges are provided to all baselines.

## 4.1 ZERO-SHOT LINK PREDICTION ON UNSEEN DOMAINS

We evaluate STAGE's zero-shot out-of-domain generalization using the E-Commerce Stores dataset with five distinct product categories, each having unique features. We train models on a union of four categories and test on the fifth held-out category, predicting customer-product interaction links.

**Results.** Figure 3 shows that STAGE consistently outperforms all baselines in zero-shot Hits@1 scores across test domains, with significant gains in the *smartphone* category, achieving a substantial 103% Hits@1 improvement against the best baseline (NBFNet-structural), with 0.51 Hits@1 for STAGE vs 0.25 for structural. Similar gains are observed in the *bed* and *refrigerator* categories, where STAGE outperforms the strongest baselines (LLM and normalized feature encodings), by margins of 40% (with a Hits@1 score of 0.44) and 33% (with a Hits@1 score of 0.59), respectively.

Table 1 shows the Hits@1 and Mean Reciprocal Rank (MRR) averaged over all held-out E-Commerce Stores. STAGE achieves the highest zero-shot Hits@1 showing a 41% improvement over the strongest baseline (normalized), with a Hits@1 of 0.46. In terms of MRR, STAGE also leads with an MRR of 0.4971, outperforming the best baseline by 29%. Interestingly, not only STAGE's outperforms the zero-shot SOTA in average, but it also exhibits lower variance across seeds (shown in both Figure 3 and Table 1), emphasizing its robustness to these zero-shot domain-generalization tasks.

## 4.2 ZERO-SHOT LINK PREDICTION WITH EXTREME DOMAIN SHIFT

To further challenge STAGE, we apply the pretrained NBFNet-STAGE on the E-Commerce Stores to zero-shot predict activities in the H&M dataset. The H&M dataset, sourced from an entirely different data provider, with distinct customers, products, features, and activity patterns, works to eliminate any potential shared predictive signals present in the E-Commerce dataset (Kechinov, 2020).

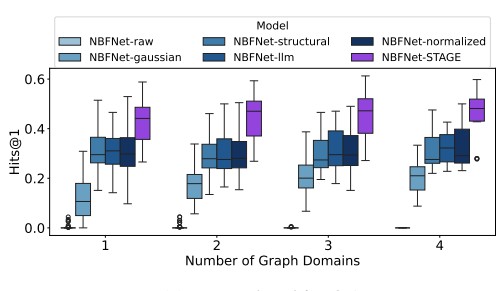 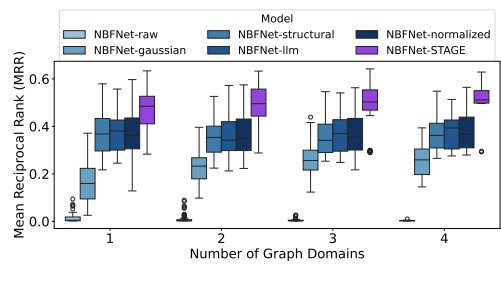

(a) Zero-shot hits@1            (b) Zero-shot MRR

Figure 4: **STAGE's zero-shot performance improves with more train domains, while this is not the case for other methods.** Zero-shot Hits@1 and MRR plots for different numbers of training graph domains. Box-plot distribution over all combinations of a fixed number of graph domains in the E-Commerce Stores Dataset and testing on the held-out domain, averaged over random seeds.

**Results. Table 1 shows the zero-shot performance on H&M of our pretrained STAGE on E-Commerce Stores is virtually identical to its performance on the held-out E-Commerce Stores (0.46 vs. 0.46 on Hits@1).** This highlights the robustness of STAGE to substantial domain shifts, as E-Commerce Stores mostly sells completely different products (household items, electronics, and shoes) from H&M (primarily clothing, very few shoes (Cuoghi, 2021)).

Compared to the baselines, STAGE achieves a relative zero-shot out-of-domain Hits@1 improvement of 102% over the best baseline (NBFNet-llm) (0.46 vs. 0.23 Hits@1). Moreover, STAGE achieves a relative Hits@1 improvement of 201% against a supervised structural method trained and tested on H&M (STRUCTURAL-SUPERVISED). These substantial gains in this stress-task highlights the ability of a pretrained STAGE to generalize to entirely new domains in zero-shot scenarios, demonstrating transfer learning capabilities that exceed that of the popular approach of encoding numerical features with LLMs. On MRR, STAGE leads with a score of 0.4703, surpassing the best baseline by 99%.

### 4.3 ZERO-SHOT NODE CLASSIFICATION ON UNSEEN DOMAINS

To confirm that our results are not specific to link prediction and E-Commerce tasks, we benchmark on a node classification task using the Pokec and Friendster datasets, where the goal is to predict the *gender* of each user. We train models on a sample of Friendster and evaluate zero-shot on Pokec. Appendix C presents details of this experiment.

**Results.** Table 2 shows that STAGE achieves a 10.3% improvement over the best baseline (and significantly lower standard deviation), including the existing node classification foundation model GraphAny Zhao et al. (2024). This indicates that STAGE effectively captures feature dependencies also in node classification tasks.

Table 2: Zero-shot test accuracy (higher is better) of STAGE and baselines on the pokec dataset, trained on a sample of Friendster. **STAGE demonstrates a gain of 10.88% in zero-shot test accuracy.** For GINE-age, models were trained and tested using only the shared *age* feature.

| Model | Accuracy ($\uparrow$) $\pm$ std |
|---|---|
| GINE-structural | 0.564±0.0466 |
| GINE-gaussian | 0.588±0.0250 |
| GINE-normalized | 0.541±0.0148 |
| GINE-llm | 0.550±0.0368 |
| GraphAny | 0.591±0.0083 |
| **GINE-STAGE (Ours)** | **0.652±0.0042** |

### 4.4 GENERALIZATION WHEN TRAINING ON MULTIPLE DOMAINS

We now examine how STAGE improves performance if the number of training domains increases.

**Results.** Figure 4a shows that STAGE's zero-shot out-of-domain Hits@1 and MRR performance consistently improves as the number of training domains increases. This trend is unique to STAGE, indicating that it is the only method capable of learning generalizable patterns across many domains, enabling it to accurately zero-shot predict customer interest in new domains.

**Discussion.** The superior performance of STAGE highlights its ability to generalize when node feature spaces vary significantly between train and test graphs. These results demonstrate that *STAGE-*

*edge-graph*s effectively encode dependencies between features and topology, making representations transferable across feature space domains. The consistent improvements show that ignoring node features or using generic embeddings can be insufficient for cross-domain generalization. Appendix E presents more results on ablation studies, confirming STAGE's effectiveness on alternative GNN backbones and ability to leverage dependencies among unseen attributes on test domains.

## 5 RELATED WORK

A more comprehensive discussion of related work can be found in Appendix G.

**Graphs Generalization under Distribution Shifts.** Several works address distribution shifts between train and test graphs over the same feature space, such as You et al. (2022); Zhu et al. (2021b), which employ learned augmentations to mitigate distribution shift in node attributes. Meanwhile, extensive research has focused on domain adaptation for GNNs (Dai et al., 2022; Li et al., 2020; Kong et al., 2022; Pei et al., 2020; Veličković et al., 2019; Wiles et al., 2022; Zhang et al., 2019; Zhu et al., 2021a), which typically assume access to data in both source and target domains. In contrast, our work tackles the more challenging scenario of zero-shot generalization to unseen attribute spaces.

**Foundation Models for Graph Data.** Developing foundation models for graph data is a growing research interest, aiming to create versatile graph models capable of generalizing across different graphs and tasks (Mao et al., 2024). Initial efforts in this direction convert attributed graphs into texts and apply an LLM (Liu et al., 2024; Chen et al., 2024b;a; Tang et al., 2024; Zhao et al., 2023; He & Hooi, 2024; Huang et al., 2023). However, while promising, this methodology risks information loss and may limit transferability (Collins et al., 2024; Gruver et al., 2024; Schwartz et al., 2024). In contrast, non-LLM approaches attempt to directly address domain transferability in the attribute space (Xia & Huang, 2024; Lachi et al., 2024; Zhao et al., 2024), or forgo the attributes entirely (Gao et al., 2023; Lee et al., 2023; Galkin et al., 2024). However, no definitive solution has emerged, and the search for a universal graph model for node feature domain shift remains an open challenge.

**Maximal Invariants and Statistical Testing.** Bell (1964) first explored the relationship between invariant and almost-invariant tests in hypothesis testing. Berk & Bickel (1968) and Berk (1970) extended Bell's approach to show that almost-invariant tests are equivalent to invariant ones under certain conditions, which are those met in our work. Later, Berk et al. (1996) explored the interplay between sufficiency and invariance in hypothesis testing by providing counterexamples that demonstrate how these concepts can differ significantly in other scenarios. More recently, Koning & Hemerik (2024) improves the efficiency of hypothesis testing under invariances for large transformation groups such as rotation or sign-flipping without resorting to sampling.

## 6 CONCLUSION

Designing graph models with the same intuitive flexibility as vision and language models is limited by the challenge of learning representations that are informative across diverse feature spaces, including new unseen concepts at test time. We tackle this challenge by proposing STAGE, a method designed for zero-shot generalization across graphs with distinct node feature spaces. STAGE provides a principled framework that learns embeddings by capturing how feature dependencies are structured within the graph topology, rather than feature values. STAGE shows strong empirical performance even under entirely new attribute spaces, unlike its natural baselines. We believe STAGE represents a meaningful advance in graph machine learning, paving the way for more universal pretained models that generalize across diverse datasets while fully leveraging node attributes.

**Limitations and Future Work.** While our method demonstrates scalability with respect to the number of nodes (Appendix F), it may encounter limitations when dealing with high-dimensional feature spaces. To address this challenge, various techniques can be explored, such as feature selection via association studies, which we identify as a promising direction for future research.

Looking ahead, several avenues for further investigation emerge. These include extending our framework to accommodate multi-dimensional and multi-modal attributes, such as text and images, integrating large language models (LLMs) into the pipeline, and evaluating its performance on diverse domains, including biomedical and geospatial networks. Pursuing these directions will be a positive step towards realizing a robust graph foundation model for attributed graphs.

**Reproducibility Statement.** The assumptions and proofs of our theoretical results can be found in Appendix A. A detailed description of how to obtain and process our datasets and of our tasks can be found in Appendix B. The configurations we use for experiments can be found in Appendix C. All our experiments were performed on an Nvidia A100 80GB using Pytorch and the PyTorch Geometric library (Fey & Lenssen, 2019).

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

# A  PROOFS AND FURTHER THEORETICAL RESULTS

## A.1  GROUPOIDS

**Definition A.1** (Groupoids)**.**  A *groupoid* $\mathcal{G}$ consists of the following elements:

1. A collection of distinct *spaces*, denoted as $\text{Spaces}(\mathcal{G})$.

2. A set of *transformations* (also called morphisms) between these spaces, denoted as $\text{Trans}(\mathcal{G})$.

3. Each transformation $f \in \text{Trans}(\mathcal{G})$ maps one space in $\text{Spaces}(\mathcal{G})$ to another space (or potentially to itself), denoted as $f : X \to Y$, where $X, Y \in \text{Spaces}(\mathcal{G})$.

4. There is a rule for combining transformations: for any two transformations $f : X \to Y$ and $g : Y \to Z$, their *composition* results in a transformation $g \circ f : X \to Z$.

5. Each space $S \in \text{Spaces}(\mathcal{G})$ has an *identity transformation* $\text{id}_S : S \to S$ that maps $S$ to itself, such that for any space $X \in \text{Spaces}(\mathcal{G})$ and any transformation $f_1 : S \to X$ and $f_2 : X \to S$, it guarantees $f_1 \circ \text{id}_S = f_1$ and $\text{id}_S \circ f_2 = f_2$.

6. Every transformation $f : X \to Y$ has a unique *inverse transformation* $f^{-1} : Y \to X$ such that $f^{-1} \circ f = \text{id}_X$ and $f \circ f^{-1} = \text{id}_Y$.

## A.2  STATISTICAL TESTS AS GRAPH REGRESSION ON FEATURE HYPERGRAPHS

To prove the result of Theorem 3.2, we will first show an intermediate result using the notion of *maximal invariants*. Let $\mathcal{G}$ be a transformation group acting on a space $\mathbb{X}$. A function $M : \mathbb{X} \to \mathbb{Z}$ is said to be maximal invariant if it is invariant to transformations of $\mathcal{G}$ and if $\forall x_1, x_2 \in \mathbb{X}$, $M(x_1) = M(x_2)$ implies $x_2 = g \circ x_1$ for some group action $g \in \mathcal{G}$, that is, if $M$ is constant on the orbits but for each orbit, it takes on a different value (Lehmann et al., 2005, pp. 214). A *maximal invariant* is a representation theory counterpart of *sufficient statistics*.

Our intermediate result will show that the feature hypergraph admits a graph representation that is a maximal invariant. But first, we need to formally define the class of invariances, which we show later is essential for STAGE's domain transferability. Since we are interested in attribute spaces of distinct domains, rather than using groups (which involve automorphisms mapping a space onto itself), we will use groupoids (Definition A.1). Groupoids generalize the concept of groups by allowing transformations between multiple spaces. In a group, all transformations map a space onto itself, while in a groupoid, transformations can map between different spaces, but must still be invertible.

**Definition A.2** (Component-wise order-preserving groupoids for features (COGF))**.**  Let $\mathbb{X}_1, \mathbb{X}_2$ be two feature spaces, both with $d$ feature dimensions. A feature transformation $f : \mathbb{X}_1 \to \mathbb{X}_2$ is said to be *component-wise order-preserving* if it can be decomposed into a set of maps $f_1, \ldots, f_d$, where each $f_i$ maps the $i$-th dimension of $\mathbb{X}_1$ to the $i$-th dimension of $\mathbb{X}_2$ and is a homomorphism that preserves the total order in $\mathbb{X}_1$, and all dimensions of both $\mathbb{X}_1$ and $\mathbb{X}_2$ have a mapping.

Given an edge endpoint features $\mathcal{E} = \{\{(\boldsymbol{x}^u, \boldsymbol{x}^v) \mid (u, v) \in E\}\}$ and a groupoid action $f$ from the COGF (Definition A.2), we define how $f$ acts on $\mathcal{E}$ as follows:

$$f(\mathcal{E}) = \{\{(f(\boldsymbol{x}^u), f(\boldsymbol{x}^v)) \mid (u, v) \in E\}\} .$$

Now, we are ready to establish the intermediate result as follows.

**Lemma A.3.**  *Given a multiset of edge endpoint features $\mathcal{E}$ and the feature hypergraph $\mathcal{F}_{\mathcal{E}}$ (Definition 3.1). There exists a parameterization $\theta^*$ for a maximally expressive hypergraph GNN encoder $M$ such that $M_{\theta^*}(\mathcal{F}_{\mathcal{E}})$ is a maximal invariant under COGFs (Definition A.2).*

*Proof.*  Let $\mathcal{V}(\mathcal{F}_{\mathcal{E}})$ be set of labeled nodes (labeled with the feature id and the order statistic position) of $\mathcal{F}_{\mathcal{E}}$, let $\mathcal{H}(\mathcal{F}_{\mathcal{E}})$ be set of hyperedges of $\mathcal{F}_{\mathcal{E}}$, and let $m(\mathcal{F}_{\mathcal{E}})$ be the number of entities from $\mathcal{E}$, which are labeled with the entire graph during creation. Given two hypergraphs $\mathcal{F}_{\mathcal{E}_1}, \mathcal{F}_{\mathcal{E}_2}$, we define $\mathcal{F}_{\mathcal{E}_1} = \mathcal{F}_{\mathcal{E}_2}$ if and only if $\mathcal{V}(\mathcal{F}_{\mathcal{E}_1}) = \mathcal{V}(\mathcal{F}_{\mathcal{E}_2})$, $\mathcal{H}(\mathcal{F}_{\mathcal{E}_1}) = \mathcal{H}(\mathcal{F}_{\mathcal{E}_2})$, and $m(\mathcal{F}_{\mathcal{E}}) = m(\mathcal{F}_{\mathcal{E}'})$. Note that since the node in the feature hypergraph are always labeled, a most expressive hypergraph GNN $M_{\theta^*}$ will ensure that $M_{\theta^*}(\mathcal{F}_{\mathcal{E}_1}) = M_{\theta^*}(\mathcal{F}_{\mathcal{E}_2})$ if and only if $\mathcal{F}_{\mathcal{E}_1} = \mathcal{F}_{\mathcal{E}_2}$.

Let $\mathcal{G}$ be the COGF (Definition A.2) and let $f \in \mathcal{G}$ be an arbitrary groupoid action of COFG. To show invariance, the goal is to show that $M_{\theta*}(\mathcal{F}_{\mathcal{E}}) = M_{\theta*}(\mathcal{F}_{f(\mathcal{E})})$ for any $\mathcal{E}$. Because $M_{\theta*}$ is most expressive, this is equivalent to showing $\mathcal{F}_{\mathcal{E}} = \mathcal{F}_{f(\mathcal{E})}$.

Let $\mathcal{V}(\mathcal{F}_{\mathcal{E}}) = \{(i, k, l)\}_{i \in [d], k \in [m], l \in \{1, 2\}}$. We first observe that since $f$ acts on individual feature values, it does not change the total number of entities. Hence, the set of hypergraph nodes remain unchanged, $\mathcal{V}(\mathcal{F}_{f(\mathcal{E})}) = \mathcal{V}(\mathcal{F}_{\mathcal{E}}) = m$.

For the edges, consider an arbitrary hyperedge $H_{uv}$ in $\mathcal{F}_{\mathcal{E}}$. Then, because $f$ is a COGF, it preserves the order statistics of all feature values. Thus, the order of the feature value $o_i(u)$ from $\mathcal{E}$ remains the the same as $o'_i(u)$ from $f(\mathcal{E})$, for all $i$ and $u$. Hence, $H_{uv}$ is also a hyperedge in $\mathcal{F}_{f(\mathcal{E})}$. Similarly, because $f$ has an inverse $f^{-1}$, we can show that for every edge $H'_{uv}$ in $\mathcal{F}_{f(\mathcal{E})}$, it is also in $\mathcal{F}_{\mathcal{E}}$ under the transformation $f^{-1}$. Thus, $\mathcal{H}(\mathcal{F}_{\mathcal{E}}) = \mathcal{H}(\mathcal{F}_{f(\mathcal{E})})$ and so $\mathcal{F}_{\mathcal{E}} = \mathcal{F}_{f(\mathcal{E})}$, and therefore $M_{\theta}(\mathcal{F}_{\mathcal{E}}) = M_{\theta}(\mathcal{F}_{f(\mathcal{E})})$.

To show maximality, let $M_{\theta*}(\mathcal{F}_{\mathcal{E}}) = M_{\theta}(\mathcal{F}_{\mathcal{E}'})$ for some $\mathcal{E}$ and $\mathcal{E}'$. Our goal is to show that $\mathcal{E}$ and $\mathcal{E}'$ are on the same orbit, i.e. there exists a $f \in \mathcal{G}$ such that $f(\mathcal{E}) = \mathcal{E}'$.

Because $M_{\theta*}$ is most expressive, we know $\mathcal{F}_{\mathcal{E}} = \mathcal{F}_{\mathcal{E}'}$. This implies that $\mathcal{V}(\mathcal{F}_{\mathcal{E}}) = \mathcal{V}(\mathcal{F}_{\mathcal{E}'})$ and $|\mathcal{H}(\mathcal{F}_{\mathcal{E}})| = |\mathcal{H}(\mathcal{F}_{\mathcal{E}'})|$. First, Let $m = |\mathcal{V}(\mathcal{F}_{\mathcal{E}})| = |\mathcal{V}(\mathcal{F}_{\mathcal{E}'})|$. And since $\mathcal{V}(\mathcal{F}_{\mathcal{E}}) = \mathcal{V}(\mathcal{F}_{\mathcal{E}'})$, we also know both $\mathcal{E}$ and $\mathcal{E}'$ must have the same number of features. Denote it $d$. In addition, because $\mathcal{H}(\mathcal{F}_{\mathcal{E}}) = \mathcal{H}(\mathcal{F}_{G'})$, we have $|E| = |E'|$. Second, pick any endpoint features $(\boldsymbol{x}^u, \boldsymbol{x}^v) \in \mathcal{E}$, and let $H_{uv} \in \mathcal{H}(\mathcal{F}_{\mathcal{E}})$ be the corresponding hyperedge. We know that $H_{uv} \in \mathcal{H}(\mathcal{F}_{\mathcal{E}'})$ as well. Hence, there exists an counterpart endpoint features $(\boldsymbol{x}^{u'}, \boldsymbol{x}^{v'}) \in \mathcal{E}'$ such that

$$\forall 1 \leq i \leq d, o_i(u) = o'_i(u') \text{ and } o_i(v) = o'_i(v'),$$

where $o_i(\cdot)$ is the order of values of $i$-th feature in $\mathcal{F}_{\mathcal{E}}$ and $o'_i(\cdot)$ the order of values of $i$-th feature in $\mathcal{F}_{\mathcal{E}'}$. Thus, we can construct a COGF groupoid action $f$ as follows:

Let $f$ be decomposed into a set of maps $f_1, \ldots, f_d$ for every feature dimension $i$. Each $f_i$ is a piecewise linear function $f_i$ defined as follows:

$$f_i(a) = \begin{cases} a - (\boldsymbol{x})_{i(0)} + (\boldsymbol{x}')_{i(0)} & \text{if } a < (\boldsymbol{x})_{i(0)} \\ (\boldsymbol{x}')_{i(k)} & \text{if } a = \boldsymbol{x}_{i(k)} \text{ for some } k : 1 \leq k \leq m'_i \\ (\boldsymbol{x}')_{i(k_0)} + \frac{(\boldsymbol{x}')_{i(k_1)} - (\boldsymbol{x}')_{i(k_0)}}{(\boldsymbol{x})_{i(k_1)} - (\boldsymbol{x})_{i(k_0)}}(a - (\boldsymbol{x})_{i(k_0)}) & \text{if } \boldsymbol{x}_{i(k_0)} < a < \boldsymbol{x}_{i(k_1)} \text{ for some} \\ & \quad k_0, k_1 : 1 \leq k_0 < k_1 \leq m'_i \\ a - (\boldsymbol{x})_{i(m'_i)} + (\boldsymbol{x}')_{i(m'_i)} & \text{if } a > (\boldsymbol{x})_{i(m'_i)} \end{cases}$$

Since each $f_i$ is a piecewise linear function that strictly increases, each of them preserves the order of feature values. And since $f$ can be decomposed into $f_i$'s, $f$ is a COGF groupoid action.

Hence, we have showed that there exists a $f$ such that $f(\mathcal{E}) = \mathcal{E}'$ which shows maximality. Hence completing the proof.

$\square$

Based on Lemma A.3, we are ready to prove that measuring dependencies of the features $(\boldsymbol{x}^u, \boldsymbol{x}^v) \in \mathcal{E}$ under COGF invariances can be defined as depending only on a most-expressive GNN encoding of the feature hypergraph $\mathcal{F}_{\mathcal{E}}$. In short, this is because any hypothesis test $T(\mathcal{E})$ that can be expressed as rank test is invariant to COGF, and any invariant function can necessarily be expressed as depending only on a maximal invariant.

**Theorem 3.2.** *Given a multiset of edge endpoint features $\mathcal{E}$, the corresponding feature hypergraph $\mathcal{F}_{\mathcal{E}}$ (Definition 3.1) and a most-expressive hypergraph GNN encoder $M_{\theta*}(\mathcal{F}_{\mathcal{E}})$, then any test $T(\mathcal{E})$ that focuses on measuring the dependence of the endpoint features of $\mathcal{E}$ has an equivalent function $h$ within the space of Multilayer Perceptrons (MLPs) that depends solely on the graph representation $M_{\theta*}(\mathcal{F}_{\mathcal{E}})$, i.e., $\exists h \in MLPs \text{ s.t. } T(\mathcal{E}) = h(M_{\theta*}(\mathcal{F}_{\mathcal{E}}))$.*

*Proof.* We first note that any test $T(\mathcal{E})$ that focuses on measuring the dependence or independence of endpoint features of $\mathcal{E}$ is necessarily a rank test that relies solely on the indices of the order statistics rather than the numerical values of the features (Bell, 1964; Berk & Bickel, 1968). As such, $T(\mathcal{E})$

is invariant to COGFs (Definition A.2). Now, we show that given $\mathcal{E}$, $\mathcal{F}_{\mathcal{E}}$, and a most expressive hypergraph GNN encoder $M_{\theta^*}$, there exists an $h$ such that $T(\mathcal{E}) = h(M_{\theta^*}(\mathcal{F}_{\mathcal{E}}))$.

For any $\mathcal{E}_1, \mathcal{E}_2$, we know that if $M_{\theta^*}(\mathcal{F}_{\mathcal{E}_1}) = M_{\theta^*}(\mathcal{F}_{\mathcal{E}_2})$, then $f(\mathcal{E}_1) = \mathcal{E}_2$ for some groupoid action $f$ in COGF (Lemma A.3). Then, because $T$ is invariant to $f$, we have that $T(\mathcal{E}_1) = T(f(\mathcal{E}_1)) = T(\mathcal{E}_2)$. Hence, each value of $M_{\theta^*}(\mathcal{F}_{\mathcal{E}})$ is associated with no more than one value of $T(\mathcal{E})$. In other words, there exists a mapping $h^*$ such that $h^*(M_{\theta^*}(\mathcal{F}_{\mathcal{E}})) = T(\mathcal{E})$.

Since MLPs are universal function approximators (Leshno et al., 1993), there exists a MLP $h$ that approximates $h^*$, i.e., $h(M_{\theta^*}(\mathcal{F}_{\mathcal{E}})) = T(\mathcal{E})$.

$\square$

### A.3 Correspondence between STAGE-edge-graphs and feature hypergraphs

For the proof of Theorem 3.3, we first prove an intermediate result, which establishes that there exists a bijective mapping between the feature hypergraph $\mathcal{F}_{\mathcal{E}}$ and the multiset of stage graphs, $\mathbb{S}_E := \{\{\boldsymbol{G}(\boldsymbol{S}^{uv}) \mid (u,v) \in E\}\}$, where each STAGE-edge-graph is equipped with unique feature ids. In the case of repeated feature values, we will show a bijective mapping to a collapsed feature hypergraph, where the nodes corresponding to the repeated feature values are collapsed into one single node, with its order $k$ being the smallest order of these repeated values. We denote by $n'_i$ the number of unique feature values of feature $i$. We note that such a collapsed feature hypergraph in the case of repeated feature values will provide a representation that is stabler than the traditional rank tests, as repeated values will translates into uncertainty or noise in the rank test results, whereas our feature hypergraph representation will remain stable.

**Lemma A.4.** *There exists a bijective mapping $\mathcal{I}$ between the multiset of STAGE-edge-graphs $\mathbb{S}_E := \{\{\boldsymbol{G}(\boldsymbol{S}^{uv}) \mid (u,v) \in E\}\}$ with unique* feature ids *and the feature hypergraph $\mathcal{F}_{\mathcal{E}}$ (Definition 3.1).*

*Proof.* Let $G = (V, E, \boldsymbol{X})$ be an input graph and let $\mathcal{E} = \{\{(\boldsymbol{x}^u, \boldsymbol{x}^v) \mid (u,v) \in E\}\}$ be the corresponding multiset of edge endpoint features. We assume that each stage graph $\boldsymbol{G}(\boldsymbol{S}^{uv}) \in \mathbb{S}_E$ has nodes labeled as follows: the node associated with $i$-th feature for the source node $u$ is labeled with $(i, 1)$, and the node associated with $i$-th feature for the target node $v$ is labeled with $(i, 2)$, for every feature $i \in [d]$. Thus, given the graph $\boldsymbol{G}(\boldsymbol{S}^{uv})$, we can recover weighted adjacency matrix $\boldsymbol{S}^{uv}$, and so there is a one-to-one mapping between them. Hence, for the following discussion, we refer to $\boldsymbol{G}(\boldsymbol{S}^{uv})$ and $\boldsymbol{S}^{uv}$ interchangeably.

We first show that, given $\mathbb{S}_E$, we can construct $\mathcal{F}_{\mathcal{E}}$.

**Construct $\mathcal{I} : \mathcal{I}(\mathbb{S}_E) = \mathcal{F}_{\mathcal{E}}$:**

We first construct the set of feature hypergraph nodes. For every feature $i$, collect the multiset $Q_{i1} = \{\{\boldsymbol{S}^{uv}_{ii}\}\}_{(u,v) \in E}$ and $Q_{i2} = \left\{\left\{\boldsymbol{S}^{uv}_{(i+d)(i+d)}\right\}\right\}_{(u,v) \in E}$ and let $Q_i = Q_{i1} \cup Q_{i2}$. In words, $Q_{i1}$ collects the i-th feature's empirical c.d.f., $\boldsymbol{S}^{uv}_{ii} = p(x^u_i) = \mathbb{P}(\mathrm{x}_i \leq x^u_i)$, of the source node $u$ of all edges. Similarly, $Q_{i2}$ collects the i-th feature's empirical c.d.f., $\boldsymbol{S}^{uv}_{(i+d)(i+d)} = p(x^v_i) = \mathbb{P}(\mathrm{x}_i \leq x^v_i)$, of the target node $v$ of all edges. Note that $Q_i$ is a multiset, so if there are multiple nodes $u$ (or $v$) with the same i-th feature value $x^u_i$ (or $x^v_i$), they will have the same empirical c.d.f. $p(x^u_i)$ (or $p(x^v_i)$), and thus $Q_i$ will record the multiplicity (number of occurrence) of such repeated c.d.f. values.

Sort the unique values in the multiset $Q_i$ in ascending order and denote the sorted sequence of unique values as $S_i = (s_1, s_2, \ldots, s_{m'_i})$ where $s_l \in Q_i$ for each $l \in [m'_i]$ where $m'_i \leq m$ is the total number of unique values for feature $i$ (if all values have multiplicity of 1, then $m'_i = m$). Denote $n_i(s_l)$ the multiplicity of the value $s_l$ in the multiset $Q_i$. Then, we can recover the feature hypergraph's nodes corresponding to the $i$-th feature as follows:

- For the smallest feature value, Ccnstruct the two nodes labeled $(i, 1, 1)$ and $(i, 1, 2)$.

- For $l \in \{2, 3, \ldots, m'_i\}$ and $s_l \in S_i$, construct the two nodes labeled as $(i, l - 1 + n_i(s_{l-1}), 1)$ and $(i, l - 1 + n_i(s_{l-1}), 2)$. $l - 1 + n_i(s_{l-1})$ is the order of the feature value $s_l$, accounting for multiplicity.

Repeating the above process for all features $i$ will recover the node set of the feature hypergraph.

Then, we reconstruct the multiset of hyperedges. Take any $\boldsymbol{S}^{uv} \in \mathbb{S}_E$. Again, for every feature $i$, we have $\boldsymbol{S}^{uv}_{ii} = p(x^u_i)$ denoting the empirical c.d.f. of the i-th feature of the source node $u$. Let $N_{iu} = \{(i, k_{i,1}, 1), \ldots, (i, k_{i,m'_i}, 1)\}$, where the $k_{i,l}$'s are the orders ($l \in [m'_i]$). $N_{iu}$ then is the subset of hypernodes for feature $i$ associated with the source node $u$ in the original edge in the input graph. Now, let $l'$ be the smallest integer in $[m'_i]$ such that $k_{i,l'} > \boldsymbol{S}^{uv}_{ii} = p(x^u_i)$, and let $k^u_i = k_{i,l'} - 1$. Then, $k^u_i$ is the order of the i-th feature of node $u$, i.e., $k^u_i = o_i(u)$.

Similarly, for every feature $i$, we have $\boldsymbol{S}^{uv}_{(i+d)(i+d)} = p(x^v_i)$, the empirical marginal c.d.f. when node $v$ is the target node of an edge. Let $N_{iv} = \{(i, k_{i,1}, 2), \ldots, (i, k_{i,m'_i}, 2)\}$. Let $l''$ be the smallest integer in $[m'_i]$ such that $k_{l''} > \boldsymbol{S}^{uv}_{(i+d)(i+d)} = p(x^v_i)$. Then, let $k^v_i = k_{l''} - 1$, and this is the order of the i-th feature of node $v$, i.e., $k^v_i = o_i(v)$.

Hence, we have recovered the hyperedge:

$$H_{uv} = \{(i, k^u_i, 1)\}_{i \in [d]} \cup \{(i, k^v_i, 2)\}_{i \in [d]}.$$

where $k^u_i$ and $k^v_i$ are defined as above.

Repeat the above process for every $\boldsymbol{S}^{uv} \in \mathbb{S}_E$, then we recover the entire multiset of hyperedges for the feature hypergraph.

**Construct $\mathcal{I}^{-1}$ : $\mathcal{I}^{-1}(\mathcal{F}_{\mathcal{E}}) = \mathbb{S}_E$.**

Given a feature hypergraph $\mathcal{F}_{\mathcal{E}}$ with $\mathcal{V}(\mathcal{F}_{\mathcal{E}})$ the set of nodes and $\mathcal{H}(\mathcal{F}_{\mathcal{E}})$ the multiset of hyperedges. Our goal is to reconstruct the multiset of STAGE-edge-graphs $\mathbb{S}_E = \{\!\{\boldsymbol{G}(\boldsymbol{S}^{uv}) \mid (u, v) \in E\}\!\}$ for some underlying edge set $E$.

Pick any hyperedge $H = \{(i, k^1_i, 1)\}_{i \in [d]} \cup \{(i, k^2_i, 2)\}_{i \in [d]} \in \mathcal{H}(\mathcal{F}_{\mathcal{E}})$, where $k^1_i = o_i(u)$ is the order of i-th feature value for some unknown node $u$ and $k^2_i = o_i(v)$ the order of i-th feature value for some unknown node $v$. We first construct the corresponding STAGE-edge-graph adjacency matrix, which we denote $\boldsymbol{S}^H$. Once $\boldsymbol{S}^H$ is obtained, then we have the STAGE-edge-graph $\boldsymbol{G}(\boldsymbol{S}^H)$.

First, we construct the diagonal entries of $\boldsymbol{S}^H$ as follows. Note that the entire hypergraph is labeled with an integer $m$, which indicate the total number of entities (nodes) in the original input graph. Hence, we can recover the marginal empirical c.d.f. of the i-th feature value of each entity. Specifically, for every feature $i$, we have $k^1_i$ from the hyperedge $H$, denoting the order of i-th feature value of the underlying source node $u$ of an edge in the original input graph. If there is another hypergraph node $(i, k', 1) \in \mathcal{V}(\mathcal{F}_{\mathcal{E}})$ such that $k' > k^1_i$, then let $n^1_i = k' - 1$. Otherwise, let $n^1_i = m$. Thus, $n^1_i$ indicates the total number of nodes in the original input graph that have the i-th feature values smaller than or equal to the i-th feature value of the current node $u$. Note that $n^1_i$ accounts for multiplicity, if there were multiple nodes having the same i-th feature value as this node. Hence, let $\boldsymbol{S}^H_{ii} = n^1_i/m$, which is equal to the marginal empirical c.d.f. of the i-th feature value of node $u$.

Similarly, for every feature $i$ we have $k^2_i$. If there is another hypergraph node $(i, k', 2) \in \mathcal{V}(\mathcal{F}_{\mathcal{E}})$ such that $k' > k^2_i$, then let $n^2_i = k' - 1$. Otherwise, let $n^2_i = m$. Let $\boldsymbol{S}^H_{(i+d)(i+d)} = n^2_i/m$. Hence, we have filled in the diagonal entries of $\boldsymbol{S}^H$.

Second, we construct the off-diagonal entries of $\boldsymbol{S}^H$. Recall that the off-diagonal entries of STAGE-edge-graph weighted adjacency matrices denote the empirical conditional probabilities between two different features (Equation (2)), either within the same source node, the same target node, or between the source and target node. Specifically, for any two features $i, j \in [d]$, $i \neq j$, the entry is

$$\boldsymbol{S}^H_{ij} = \mathbb{P}_{A \sim \text{Unif}(V)}\big(\mathrm{x}^A_i \le x^u_i \mid \mathrm{x}^A_j \le x^u_j\big)$$

$$\boldsymbol{S}^H_{i(j+d)} = \mathbb{P}_{(A,B) \sim \text{Unif}(E)}\big(\mathrm{x}^A_i \le x^u_i \mid \mathrm{x}^B_j \le x^v_j\big)$$

$$\boldsymbol{S}^H_{(i+d)j} = \mathbb{P}_{(A,B) \sim \text{Unif}(E)}\big(\mathrm{x}^B_i \le x^v_i \mid \mathrm{x}^A_j \le x^u_j\big)$$

$$\boldsymbol{S}^H_{(i+d)(j+d)} = \mathbb{P}_{B \sim \text{Unif}(V)}\big(\mathrm{x}^B_i \le x^v_i \mid \mathrm{x}^B_j \le x^v_j\big)$$

where $(u, v)$ is the edge in the input graph corresponding to the hyperedge $H$.

We can compute these entries of $\boldsymbol{S}^H$ as follows. First, given any hyperedge $H' \in \mathcal{V}(\mathcal{F}_{\mathcal{E}})$, denote $K^d_{H'}(i)$ for any $i \in [d]$ and $r \in \{1, 2\}$ such that $(i, K^d_{H'}(i), r) \in H'$. Then, regarding our particular

hyperedge $H$ of interest, for every pair of features $i, j \in [d]$ with $i \neq j$, we can obtain $n_i^1, n_j^1, n_i^2$, and $n_j^2$ as defined previously. Recall that $n_i^d$ is the number of feature values of $i$-th feature that are smaller than or equal to the current $i$-th feature value captured by $H$, for both the source node ($r = 1$) or the target node ($r = 2$).

For the entry $\boldsymbol{S}_{ij}^H$ and $\boldsymbol{S}_{(i+d)(j+d)}^H$, they capture inner-node feature dependencies, and we notice that the empirical conditional probabilities are defined w.r.t. random nodes sampled uniformly from the set of all nodes. Hence, we can compute these two entries as follows:

$$\boldsymbol{S}_{ij}^H = \min\{1, n_i^1/n_j^1\}$$
$$\boldsymbol{S}_{(i+d)(j+d)}^H = \min\{1, n_i^2/n_j^2\}.$$

To compute the entries for $\boldsymbol{S}_{i(j+d)}^H$ and $\boldsymbol{S}_{(i+d)j}^H$, we note that the random nodes A, B are uniformly sampled from the set of edges $E$. To do so, we first define the two subsets of hyperedges $\mathcal{H}_j^1$ and $\mathcal{H}_j^2$ as follows:

$$\mathcal{H}_j^1 := \{K_{H'}^1(j) \leq n_j^1 \mid H' \in \mathcal{H}(\mathcal{F}_\mathcal{E})\}$$
$$\mathcal{H}_j^2 := \{K_{H'}^2(j) \leq n_j^2 \mid H' \in \mathcal{H}(\mathcal{F}_\mathcal{E})\}.$$

In other words, $\mathcal{H}_j^1$ is the subset of hyperedges whose node, $(i, K_{H'}^1(j), 1)$, has an order $K_{H'}^1(j)$ that is smaller than or equal to the order of the counterpart node of the current hyperedge $H$. Vice versa for $\mathcal{H}_j^2$. Hence, we have

$$|\mathcal{H}_j^1|/|\mathcal{H}(\mathcal{F}_\mathcal{E})| = \mathbb{P}_{(A,B) \in \text{Unif}(E)}(x_j^A \leq x_j^u)$$
$$|\mathcal{H}_j^2|/|\mathcal{H}(\mathcal{F}_\mathcal{E})| = \mathbb{P}_{(A,B) \in \text{Unif}(E)}(x_j^B \leq x_j^v).$$

Then, we define the next two subsets $\mathcal{H}_{i|j}^{1|2}$ and $\mathcal{H}_{i|j}^{2|1}$ as follows:

$$\mathcal{H}_{i|j}^{1|2} := \{K_{H'}^1(i) \leq n_i^1 \mid H' \in \mathcal{H}_j^2\}$$
$$\mathcal{H}_{i|j}^{2|1} := \{K_{H'}^2(i) \leq n_i^2 \mid H' \in \mathcal{H}_j^1\}$$

These two subsets help us effectively computes the empirical conditional probabilities. Namely, now we have

$$|\mathcal{H}_{i|j}^{1|2}|/|\mathcal{H}_j^2| = \mathbb{P}_{(A,B) \in \text{Unif}(E)}(x_i^A \leq x_i^u \mid x_j^B \leq x_j^v)$$
$$|\mathcal{H}_{i|j}^{2|1}|/|\mathcal{H}_j^1| = \mathbb{P}_{(A,B) \in \text{Unif}(E)}(x_i^B \leq x_i^v \mid x_j^A \leq x_j^u)$$

Thus, we set the adjacency matrix entries for inter-node dependencies to

$$\boldsymbol{S}_{i(j+d)}^H = |\mathcal{H}_{i|j}^{1|2}|/|\mathcal{H}_j^2|$$
$$\boldsymbol{S}_{(i+d)j}^H = |\mathcal{H}_{i|j}^{2|1}|/|\mathcal{H}_j^2|$$

Now that we have constructed a mapping $\mathcal{I}$ mapping $\mathbb{S}_E$ to $\mathcal{F}_\mathcal{E}$, and another mapping $\mathcal{I}^{-1}$ mapping $\mathcal{F}_\mathcal{E}$ to $\mathbb{S}_E$, we now want to check that they are valid bijections. To show this, we show that $\mathcal{I}^{-1} \circ \mathcal{I} = \text{Identity}$, and $\mathcal{I} \circ \mathcal{I}^{-1} = \text{Identity}$.

**Show that $\mathcal{I}^{-1} \circ \mathcal{I} = \textbf{Identity}$**

Let $\mathbb{S}_E$ be an arbitrary multiset of STAGE-edge-graphs. Let $\mathcal{F}' = \mathcal{I}(\mathbb{S}_E)$ and $\mathbb{S}'' = \mathcal{I}^{-1}(\mathcal{F}') = \mathcal{I}^{-1}(\mathcal{I}(\mathbb{S}_E))$. First, we observe that the mapping $\mathcal{I}$ transforms each element $\boldsymbol{G}(\boldsymbol{S}^{uv}) \in \mathbb{S}_E$ to one hyperedge $H' \in F'$. Similarly, the mapping $\mathcal{I}^{-1}$ transforms each hyperedge $H' \in F'$ to one STAGE-edge-graph $\boldsymbol{G}'' \in \mathbb{S}''$. Hence, as long as we show that, for any $\boldsymbol{G}(\boldsymbol{S}^{uv}) \in \mathbb{S}_E$, the composed

transformation $\mathcal{I}^{-1} \circ \mathcal{I}$ produces a STAGE-edge-graph $\boldsymbol{G}''$ such that $\boldsymbol{G}(\boldsymbol{S}^{uv}) = \boldsymbol{G}''$, we can conclude $\mathcal{I}^{-1} \circ \mathcal{I} = $ Identity.

To observe this, we first note that $\boldsymbol{G}''$ has the same set of labeled nodes with $\boldsymbol{G}$, and that each node $(i, r), i \in [d], r \in \{1, 2\}$ has the same empirical marginal c.d.f. values. Similarly, between any two nodes $(i_1, r_1)$ and $(i_2, r_2)$, $\boldsymbol{G}$ and $\boldsymbol{G}''$ will have the same edge attribute for the edge $((i_1, r_1), (i_2, r_2))$, which corresponds to the empirical conditional probabilities between features $i_1$ and $i_2$ and between node placement in the original edge (source or target) $r_1$ and $r_2$. Thus, $\boldsymbol{G} = \boldsymbol{G}''$.

**Show that $\mathcal{I} \circ \mathcal{I}^{-1} = $ Identity**

Let $\mathcal{F}_{\mathcal{E}}$ be an arbitrary feature hypergraph. Let $\mathbb{S}' = \mathcal{I}^{-1}(\mathcal{F}_{\mathcal{E}})$ and $\mathcal{F}'' = \mathcal{I}(\mathbb{S}')$. Similarly, as long as we show that, for any hyperedge $H \in \mathcal{F}_{\mathcal{E}}$, the composed transformation $\mathcal{I} \circ \mathcal{I}^{-1}$ produces a hypergraph $H''$ such that $H = H''$, we can conclude that $\mathcal{I} \circ \mathcal{I}^{-1} = $ Identity.

To observe this, we note that every hyperedge $(i, k, r) \in H$, where $i \in [d], 1 \leq k \leq m_i', r \in \{1, 2\}$, will be recovered in $H''$. This is because each $(i, k, r) \in H$ corresponds to a unique labeled node $(i, r)$ in the STAGE-edge-graph $\mathbb{G}'$, which will be used to construct a node $(i, k'', r)$ in $H''$ under the mapping $\mathcal{I}$. In terms of the order $k$, the mapping $\mathcal{I}^{-1}$ will converts it into the marginal empirical c.d.f. value, which is treated as the attribute of node $(i, r)$ in the STAGE-edge-graph $\mathbb{G}'$. The mapping $\mathcal{I}$, on the other hand, will convert this marginal empirical c.d.f. value into the order $k''$ for the node $(i, k'', r)$ in $H''$, guaranteeing $k'' = k$. Thus, every node $(i, k, r)$ that is in $H$ is also in $H''$, and there will be no additional nodes created for $H''$. Hence, $H = H''$ for every hyperedge $H \in \mathcal{F}_{\mathcal{E}}$, and thus $\mathcal{I} \circ \mathcal{I}^{-1} = $ Identity.

In conclusion, we have shown two mappings, $\mathcal{I}$ and $\mathcal{I}^{-1}$, and have shown that they are the inverse transformation of each other. Hence, $\mathcal{I}$ is a bijective mapping between the multiset of STAGE-edge-graphs and feature hypergraph. $\qquad\square$

Given the bijective mapping in Lemma A.4 between the multiset of STAGE-edge-graphs with unique feature identifiers and the feature hypergraph, and the fact that the feature hypergraph allows for a maximal invariant graph representation (Lemma A.3), it follows that the set of STAGE-edge-graphs can also yield a maximal invariant representation of the original input graph. This observation is formalized as below, which is our second theoretical contribution:

**Theorem 3.3.** *Given the endpoint features $\mathcal{E}$ (Definition 3.1) of a graph $G = (V, E, \boldsymbol{X})$, there exists an optimal parameterization $\theta_g^*, \theta_s^*$ for a most expressive GNN encoder $M^g$ and a most-expressive multiset encoder $M^s$, respectively, such that $M_{\theta_s^*, \theta_g^*}(G) := M_{\theta_s^*}^s\left(\left\{\left\{M_{\theta_g^*}^g(\boldsymbol{G}(\boldsymbol{S}^{uv})) : (u, v) \in E\right\}\right\}\right)$ such that any test $T(\mathcal{E})$ that measures the dependence of $\mathcal{E}$'s endpoint features has an equivalent function $h$ within the space of Multilayer Perceptrons (MLPs) that depends solely on the graph representation $M_{\theta_s^*, \theta_g^*}(G)$, i.e., $\exists h \in$ MLPs s.t. $T(\mathcal{E}) = h(M_{\theta_s^*, \theta_g^*}(G))$.*

*Proof.* To show invariance, let $G_1 = (V, E, \boldsymbol{X}_2)$ and $G_2 = (V, E, \boldsymbol{X}_2)$ be two graphs such that $f(\boldsymbol{X}_1) = \boldsymbol{X}_2$ for some groupoid action $f$ in the COGF. Let $\mathcal{E}_1$ and $\mathcal{E}_2$ be the corresponding edge endpoint features respectively, from which we have $f(\mathcal{E}_1) = \mathcal{E}_2$. Let $\mathbb{S}_{1E} = \{\{\boldsymbol{G}(\boldsymbol{S}_1^{uv}) \mid (u, v) \in E\}\}$ and $\mathbb{S}_{2E} = \{\{\boldsymbol{G}(\boldsymbol{S}_2^{uv}) \mid (u, v) \in E\}\}$ be the corresponding STAGE-edge-graphs respectively.

Since $f(\mathcal{E}_1) = \mathcal{E}_2$, and the feature hypergraph is invariant to COGF (shown in the proof for Lemma A.3), we have $\mathcal{F}_{\mathcal{E}_1} = \mathcal{F}_{\mathcal{E}_2}$. And since there is a one-to-one mapping between the multiset of STAGE-edge-graphs and the feature hypergraph, we have $\mathbb{S}_{1E} = \mathbb{S}_{2E}$. Hence,

$$\left\{\left\{M_{\theta_g^*}(\boldsymbol{S}_1^{uv}) \mid (u, v) \in E\right\}\right\} = \left\{\left\{M_{\theta_g^*}(\boldsymbol{S})\right\}\right\}_{\boldsymbol{S} \in \mathbb{S}_{1E}}$$

$$= \left\{\left\{M_{\theta_g^*}(\boldsymbol{S})\right\}\right\}_{\boldsymbol{S} \in \mathbb{S}_{2E}} = \left\{\left\{M_{\theta_g^*}(\boldsymbol{S}_2^{uv}) \mid (u, v) \in E\right\}\right\}.$$

As a result,

$$M_{\theta_s^*, \theta_g^*}(G_1) = M_{\theta_s^*}^s\left(\left\{\left\{M_{\theta_g^*}^g(\boldsymbol{G}(\boldsymbol{S}_1^{uv})) \mid (u, v) \in E\right\}\right\}\right)$$

$$= M_{\theta_s^*}^s\left(\left\{\left\{M_{\theta_g^*}^g(\boldsymbol{G}(\boldsymbol{S}_2^{uv})) \mid (u, v) \in E\right\}\right\}\right) = M_{\theta_s^*, \theta_g^*}(G_2).$$

To show maximality, Let $G_1$ and $G_2$ be two graphs such that $M_{\theta_s^*, \theta_g^*}(G_1) = M_{\theta_s^*, \theta_g^*}(G_2)$. Then, because $M_{\theta_s^*}^s$ is a most expressive multiset encoder, we have that

$$M_{\theta_s^*}^s(\{\{ M_{\theta_g^*}^g(\boldsymbol{G}(\boldsymbol{S}_1^{uv})) \mid (u,v) \in E \}\}) = M_{\theta_s^*}^s(\{\{ M_{\theta_g^*}^g(\boldsymbol{G}(\boldsymbol{S}_2^{uv})) \mid (u,v) \in E \}\}).$$

Again, since $M_{\theta_g^*}^g$ is a most expressive GNN, we have

$$\mathcal{S}_{1E} = \{\{ \boldsymbol{G}(\boldsymbol{S}_1^{uv}) \mid (u,v) \in E \}\} = \{\{ \boldsymbol{G}(\boldsymbol{S}_2^{uv}) \mid (u,v) \in E \}\} = \mathcal{S}_{2E}.$$

This implies that the feature hypergraphs $\mathcal{F}_{\mathcal{E}_1}$ and $\mathcal{F}_{\mathcal{E}_2}$ are the same, $\mathcal{F}_{\mathcal{E}_1} = \mathcal{F}_{\mathcal{E}_2}$ due to the bijective mapping between multisets of STAGE-edge-graphs and feature hypergraphs. And as has been shown in the proof of Lemma A.3, this implies there exists a groupoid action $f$ in COGF such that $f(\mathcal{E}_1) = \mathcal{E}_2$. Hence, we have shown that $M_{\theta_s^*, \theta_g^*}(G)$ is a maximal invariant representation w.r.t. COGF.

Thus, similar to the proof of Theorem 3.2, there exists a MLP $h$ such that for any test $T(\mathcal{E})$, we have

$$T(\mathcal{E}) = h(M_{\theta_s^*, \theta_g^*}(G)).$$

$\square$

## A.4 COGG INVARIANCES

**Definition A.5** (Component-wise order-preserving groupoid for graphs (COGG)). Denote $\mathbb{X}$ the space of node features with $d \geq 1$ dimensions, and $\mathbb{G}(\mathbb{X})$ the space of attributed graphs with feature space $\mathbb{X}$ and $m \geq 2$ entities. A graph transformation $g : \mathbb{G}(\mathbb{X}_1) \to \mathbb{G}(\mathbb{X}_2)$ of two feature spaces $\mathbb{X}_1$ and $\mathbb{X}_2$ is said to be a groupoid action of the *component-wise order-preserving groupoid for graphs* if it can be decomposed into a permutation of node identities $g_{\text{node}} : V \to V$ and a transformation of node features $g_{\text{feature}} : \mathbb{X}_1 \to \mathbb{X}_2$ satisfying the following. Given $G_1 = (V, E_1, \boldsymbol{X}_1) \in \mathbb{G}(\mathbb{X}_1)$ and $G_2 = (V, E_2, \boldsymbol{X}_2) \in \mathbb{G}(\mathbb{X}_2)$ with $g(G_1) = G_2$,

- $\forall u, v \in V, (u, v) \in E_1 \iff (g_{\text{node}}(u), g_{\text{node}}(v)) \in E_2$ .

- $g_{\text{feature}}$ is a COGF (Definition A.2) except for any $i \in [d]$, the $i$-th component $g_{\text{feature},i}$ may map the $i$-th dimension of $\mathbb{X}_1$ to a different dimension of $\mathbb{X}_2$, while maintaining a one-to-one correspondence between all dimensions of $\mathbb{X}_1$ and $\mathbb{X}_2$.

## A.5 STAGE AS A COGG INVARIANT REPRESENTATION

**Theorem 3.4.** *STAGE (Section 2) is invariant to COGGs (Definition A.5).*

*Proof.* Given a graph $G = (V, E, \boldsymbol{X})$, a STAGE model $M$ applies two instances of equivariant GNNs, an intra-edge GNN and an inter-edge one, to process the input graph. Denote the intra-edge GNN $M_1$ and the inter-edge GNN $M_2$. The intra-edge GNN $M_1$ is applied onto $\mathbb{S}_E := \{\{ \boldsymbol{G}(\boldsymbol{S}^{uv}) \mid (u,v) \in E \}\}$, the set of STAGE-edge-graphs, to produce edge-leve embeddings:

$$\boldsymbol{r}^{uv} = M_1(\boldsymbol{G}(\boldsymbol{S}^{uv})), \forall (u,v) \in E$$

and the inter-edge GNN $M_2$ takes the edge-level embeddings as the edge attributes onto the original graph, i.e., making a $G' = (V, E, \{\{ r_{(u,v) \in E}^{uv} \}\})$ to produce a final graph representation:

$$M(G) = M_2(G') = M_2((V, E, \{\{ r^{uv} \}\}_{(u,v) \in E}))$$

Now, consider a train graph $G_{\text{tr}} = (V_{\text{tr}}, E_{\text{tr}}, \boldsymbol{X}_{\text{tr}})$ with $\mathcal{E}_{\text{tr}}$ and a test graph $G_{\text{te}} = (V_{\text{te}}, E_{\text{te}}, \boldsymbol{X}_{\text{te}})$ such that there exists a groupoid action $g$ in the COGG (Definition A.5) satisfying $g(G_{\text{tr}}) = G_{\text{te}}$. As per Definition A.5, $g$ is composed of a node identity permutation $g_{\text{node}}$ and a feature transformation $g_{\text{feature}}$.

We first note that the multiset $\{\{ r^{uv} \}\}_{(u,v) \in E}$ is invariant to node identity permutation $g_{\text{node}}$ because a multiset is invariant to the permutation of its elements. Since the inter-edge GNN $M_2$ is an equivariant GNN, we have that

$$M(g_{\text{node}}(G_{\text{tr}})) = M((g_{\text{node}}(V_{\text{tr}}), g_{\text{node}}(E_{\text{tr}}), \{\{ r^{uv} \}\}_{(u,v) \in g_{\text{node}}(E_{\text{tr}})})) = M(G_{\text{tr}}).$$

Hence, as long as we can show that the graph representation given by $M$ is also invariant under $g_{\text{feature}}$, then we together we can show that $M$ is invariant to our groupoid action $g$, and that $M(G_{\text{tr}}) = M(G_{\text{te}})$.

To proceed, we first note that the groupoid action $g_{\text{feature}}$, when applied to an attributed graph $G$, can be expressed as $g_{\text{feature}}(G) = (V, E, g_{\text{feature}}(\boldsymbol{X}))$, because the feature transformation only acts on the node features but leaves the graph structure unchanged. Hence, when applying the inner-edge GNN $M_1$ to the multiset of STAGE-edge-graphs of a transformed input graph $g_{\text{feature}}(G)$, we write $M_1(g_{\text{feature}}(\boldsymbol{G}(\boldsymbol{S}^{uv})))$, for all $(u,v) \in E$.

Now, all we need to show is that the intra-edge GNN $M_1$ produces a multiset of STAGE-edge-graph representations that is invariant under the feature transformation $g_{\text{feature}}$, i.e., $\{\!\{M_1(\boldsymbol{G}(\boldsymbol{S}^{uv}))\}\!\}_{(u,v)\in E} = \{\!\{M_1(g_{\text{feature}}(\boldsymbol{G}(\boldsymbol{S}^{uv})))\}\!\}_{(u,v)\in E}$. Since $g_{\text{feature}}$ is COGF (Definition A.2) except it may map different training feature dimensions of $\boldsymbol{X}_{\text{tr}}$ to different feature dimensions of $\boldsymbol{X}_{\text{te}}$, we can therefore further decompose it into two different components: $h$ and $f$ with $g = h \circ f$, where $h$ is a mapping that permutes feature dimensions, and $f$ is a COGF.

In Theorem 3.3, we have shown that a most expressive GNN applied to a STAGE-edge-graph $\boldsymbol{G}(\boldsymbol{S}^{uv})$ equipped with feature ids (which are the nodes ids in the STAGE-edge-graph because nodes correspond to feature dimensions) produces maximal invariant representation under COGF. Hence, this implies that the intra-edge GNN $M_1$, when applied to each STAGE-edge-graph, without unique node ids, will produce an invariant representation to the COGF $f$. Namely, for all $(u,v) \in E$,

$$M_1(f(\boldsymbol{G}(\boldsymbol{S}^{uv}))) = M_1(\boldsymbol{G}(f(\boldsymbol{S}^{uv}))) = M_1(\boldsymbol{G}(\boldsymbol{S}^{uv})).$$

Note that $f(\boldsymbol{G}(\boldsymbol{S}^{uv})) = \boldsymbol{G}(f(\boldsymbol{S}^{uv}))$ because $f$ acts on the node and edge attributes in $\boldsymbol{G}(\boldsymbol{S}^{uv})$ (which are derived from the feature values), but preserve the graph structure.

On the other hand, once the node ids in STAGE-edge-graph $\boldsymbol{G}(\boldsymbol{S}^{uv})$ is dropped, because $M_1$ is an equivariant GNN, we also have that the $M_1$'s output representations are invariant to permutations of the feature dimensions, which corresponds to the permutations of node ids in the STAGE-edge-graph. Namely, for all $(u,v) \in E$,

$$M_1(h(\boldsymbol{G}(\boldsymbol{S}^{uv}))) = M_1(\boldsymbol{G}(\boldsymbol{S}^{uv})).$$

Hence, together we have that for any $(u,v) \in E$,

$$M_1(g_{\text{feature}}(\boldsymbol{G}(\boldsymbol{S}^{uv}))) = M_1(h \circ f(\boldsymbol{G}(\boldsymbol{S}^{uv}))) = M_1(h(\boldsymbol{G}(\boldsymbol{S}^{uv}))) = M_1(\boldsymbol{G}(\boldsymbol{S}^{uv})),$$

Thus completing the proof.

$\square$

## B  DATASET CONSTRUCTION

Here we describe how we construct the E-Commerce Category Dataset, the H&M Dataset, and the Social Network Datasets (Friendster and Pokec).

### B.1  E-COMMERCE CATEGORY DATASET

To test the model's generalization to new input feature spaces, we consider a dataset of E-Commerce users and products (Kechinov, 2020). There are 29,228,809 different product categories, such as smartphones, shoes, and computers. We select a subset of the most popular product categories and form an input graph from the products under each category and their respective connected users. At test time, we hold out an entirely different graph containing unseen products, *from new unseen categories* and associated users, and test the zero-shot (i.e., frozen model) performance on the test data. In this dataset, we focus on the single task of predicting links between users and products, with links indicating a user purchasing/viewing/carting/uncarting a product.

However, all categories originally share the same features. To ensure that the graph domains we build have different feature types, we use GPT-4 to retrieve information specific to each category. Specifically, the information retrieval process involves prompting GPT-4 with the following content:

```
"According to the following information regarding an E-Commerce
    purchase, give information about the product in the following
    asked format."
"First, the product is purchased at time: " + row["event_time"] +
    "."
"Second, the category of the product is " + row["category_code"] +
    "."
"Third, the brand of the product is " + row["brand"] + "."
"Last, the price of the product is " + str(row["price"]) + "."
"Please provide information about the product in the following json
    format."
"{json_prototype}"
```

The JSON prototype is different for different categories, and contains features that are specific for the category being prompted. That is, the JSON prototype for smartphones contains, for instance, features like *display type*, which is not a feature for shoes, containing instead features such as *ankle height*. In the following, we report the JSON prototype for all categories.

**bed**

```
{
    "type": <select from ['Twin', 'Twin XL', 'Full', 'Queen', 'King
        ', 'California King']>,
    "material": <select from ['Wood', 'Metal', 'Upholstered', '
        Bamboo', 'Particle Board', 'Composite']>,
    "bed_frame_included": <select from ['True', 'False']>,
    "headboard_included": <select from ['True', 'False']>,
    "footboard_included": <select from ['True', 'False']>,
    "mattress_included": <select from ['True', 'False']>,
    "box_spring_required": <select from ['True', 'False']>,
    "weight_capacity_lbs": <give int in lbs>,
    "bed_size_length_inches": <give float in inches>,
    "bed_size_width_inches": <give float in inches>,
    "bed_size_height_inches": <give float in inches>
}
```

**desktop**

```
{
    "processor_type": <select from ['Intel Core i3', 'Intel Core i5
        ', 'Intel Core i7', 'Intel Core i9', 'AMD Ryzen 3', 'AMD
        Ryzen 5', 'AMD Ryzen 7', 'AMD Ryzen 9', 'Apple M1', 'ARM
        other']>,
    "ram_gb": <give int>,
    "storage_type_hdd_size_gb": <give int>,
    "storage_type_ssd_size_gb": <give int>,
    "storage_type_hybrid_size_gb": <give int>,
    "graphics_card": <select from ['NVIDIA GeForce GTX 1660', '
        NVIDIA GeForce RTX 2060', 'NVIDIA GeForce RTX 2070', 'NVIDIA
         GeForce RTX 2080', 'AMD Radeon RX 570', 'AMD Radeon RX
        580', 'AMD Radeon RX 590', 'AMD Radeon RX 5700', 'AMD Radeon
         RX 5700 XT']>,
    "operating_system": <select from ['Windows 10', 'macOS', 'Linux
        Ubuntu', 'Linux Fedora', 'Linux Mint', 'Debian', 'FreeBSD
        ']>,
    "power_supply_watts": <give int>,
    "cooling_system": <select from ['Air cooling', 'Liquid cooling
        ', 'Passive cooling']>,
```

```
        "has_bluetooth": <select from ['True', 'False']>
    }
```

### refrigerators

```
    {
        "energy_rating": <select from ['A+++', 'A++', 'A+', 'A', 'B', '
            C']>,
        "capacity_liters": <give int>,
        "refrigerator_type": <select from ['Top Freezer', 'Bottom
            Freezer', 'Side-by-Side', 'French Door', 'Mini Fridge', '
            Commercial']>,
        "defrost_type": <select from ['Manual', 'Frost Free', '
            Automatic Defrost']>,
        "has_ice_maker": <select from ['True', 'False']>,
        "has_water_dispenser": <select from ['True', 'False']>,
        "has_smart_technology": <select from ['True', 'False']>,
        "is_energy_efficient": <select from ['True', 'False']>,
        "height_cm": <give float>,
        "width_cm": <give float>,
        "depth_cm": <give float>
    }
```

### smartphone

```
    {
        "display_type": <select from ['OLED', 'LCD']>,
        "display_size": <give float in inches>,
        "display_resolution": <give int in pixels>,
        "processor_type": <give string>,
        "ram": <give int in GB>,
        "storage_options": <give int in GB>,
        "rear_camera_primary_resolution": <give int in MP>,
        "front_camera_resolution": <give int in MP>,
        "operating_system": <select from ['Android', 'iOS', 'HarmonyOS
            ', 'KaiOS', 'Tizen', 'Ubuntu Touch', 'PureOS', 'Sailfish OS
            ', 'Plasma Mobile']>,
        "Battery_capacity": <give int in mAh>,
        "Has_gps":  <select from ['True', 'False']>,
        "has_nfc":  <select from ['True', 'False']>
    }
```

### shoes

```
    {
        "type": <select from ['Running', 'Casual', 'Formal', 'Sports',
            'Boots', 'Sandals', 'Slippers', 'Hiking', 'Dress', 'Work', '
            Safety']>,
        "material": <select from ['Leather', 'Synthetic', 'Textile', '
            Rubber', 'Canvas', 'Mesh', 'Suede', 'Patent Leather', '
            Nubuck', 'Faux Leather']>,
        "color": <give string>,
        "size": <give float in UK sizes>,
        "gender": <select from ['Men', 'Women', 'Unisex', 'Children', '
            Infants']>,
```

```
      "closure_type": <select from ['Laces', 'Velcro', 'Slip-on', '
          Buckle', 'Zip', 'Hook and Loop', 'None']>,
      "sole_material": <select from ['Rubber', 'Synthetic', 'PVC', '
          EVA', 'Leather', 'TPU (Thermoplastic Polyurethane)', 'TPR (
          Thermoplastic Rubber)']>,
      "water_resistant": <select from ['True', 'False']>,
      "ankle_height": <select from ['Low-top', 'Mid-top', 'High-top',
           'Over the ankle']>,
      "breathability": <select from ['High', 'Medium', 'Low']>,
      "weight": <give float in grams>,
      "origin_country": <give string>,
      "seasonality": <select from ['All-season', 'Summer', 'Winter',
          'Rainy', 'Spring', 'Autumn']>,
      "eco_friendly": <select from ['True', 'False']>
  }
```

After extracting features of different numbers for all categories, we also append the original two shared features of all products (*price, brand*) that are considered to have a different distribution across categories, forming the following dataset statistics.

| Category | Number of Nodes | Number of Edges | Average Degree | Num Features |
|----------|-----------------|-----------------|----------------|--------------|
| **bed** | 4044 | 25788 | 6.38 | 13 |
| **desktop** | 3011 | 37450 | 12.44 | 12 |
| **refrigerators** | 2985 | 33520 | 11.23 | 13 |
| **smartphone** | 3391 | 31970 | 9.43 | 14 |
| **shoes** | 4032 | 54890 | 13.62 | 16 |

Table 3: Statistics of E-Commerce Categories

### B.2 H&M Dataset

H&M has 106K products, sharing the same 25 features, and 1.37M customers, sharing the same 7 features. We sampled the interaction between the most popular 830 products and 830 customers based on their node degrees. We discarded 14 product features since 12 of them are repetitive (e.g. *perceived_colour_value_id* is just a one-to-one mapping of *perceived_colour_value_name*), 1 of them is the *detail_desc* an English sentence that connects the other features, and 1 of them is the *article_id* serving as the identifier of each product. We also discarded 4 user features: *customer_id* as the identifier, *FN* and *Active* due to too many missing values (65% and 66% respectively), and *postal_code* that is overdispersed.

After picking the largest connected component of the graph formed by the 830 products and 830 users, we construct this dataset to have 77080 edges, 1580 nodes with an average degree 48.78, and 11 features for each product node and 3 features for each user node. The product features are: *product_type_name*, *product_group_name*, *graphical_appearance_name*, *colour_group_name*, *perceived_colour_value_name*, *perceived_colour_master_name*, *department_name*, *index_name*, *index_group_name*, *section_name*, *garment_group_name*. The user features are: *club_member_status*, *fashion_news_frequency*, *age*.

### B.3 Social Network Datasets (Friendster and Pokec)

The original Pokec social network dataset contains 1632803 nodes and 30622564 edges and each node has 58 features. However, 54 of them are difficult to encode either because they are random texts input by the user or because there is no straightforward way to turn the features into totally ordered ones. We first filtered out the nodes that contain invalid features and then sample the most popular 150 female and male nodes each before picking the largest connected components of the graph formed by the popular nodes.

Table 4: Comparison of statistics between the Pokec and Friendster social network datasets after filtering and sampling nodes.

| Statistics | Pokec | Friendster |
|---|---|---|
| Number of nodes | 283 | 1392 |
| Number of edges | 2084 | 3322 |
| Number of node features | 4 | 5 |
| Features | public, completion_percentage, region, age | age, interest, occupation, music, tv |
| Average degree | 7.363957597173145 | 2.3864942528735633 |
| Minimum degree | 1 | 1 |
| Maximum degree | 29 | 12 |
| Lowest degrees | [1, 2, 3, 4, 5] | [1, 2, 3, 4, 5] |
| # Nodes with lowest degrees | [35, 31, 25, 21, 14] | [516, 404, 213, 113, 62] |
| Label 0 Ratio | 50.88% | 46.84% |
| Label 1 Ratio | 49.12% | 53.16% |

The original Friendster social network dataset contains 43880 nodes and 145407 edges and each node has 644 features. However, most of the features are binary, which is inefficient for STAGE to encode (i.e. will need 644*2 nodes in each STAGE feature-edge graph). We find out the features are in the format of a meta feature (e.g. occupation) followed by a more detailed feature (e.g. writer). Therefore, we turned the binary features that share the same meta feature into a multicategorical feature. We then filtered out the nodes that have only one active binary feature under each meta feature (otherwise the multi-category does not make sense) and pick the largest connected components of the graph formed by the these nodes.

In the end, the statistics of Pokec and Friendster datasets are available in Table 4.

## C  EXPERIMENT DETAILS

For Figure 3, Table 1, and Figure 4 We use the default NBFNet-PyG configuration for the inductive WN18RR dataset (Zhu et al., 2021c), except for a few specific parameters. The input dimension for the node feature is set to 256, and the model includes six hidden layers with dimensions [256, 256, 256, 256, 256, 256], making a total of seven layers. For STAGE, we use 1 layer of GINEConv (Brossard et al., 2021) for the GNN on *STAGE-edge-graph*, which produces an edge representation of dimension 256. We also append an extra p_value to each edge in the *STAGE-edge-graph* for expressivity. All model are trained with a batch size of 32 over 30 epochs.

For Figure 3, Figure 4, and the E-Commerce columns of Table 1 we average over seeds 0, 1, 2. For the H&M columns of Table 1, we average over seeds 1024, 1025, 1026.

For Table 2, we average over seeds 32, 33, and 34 using the following configuration. The input feature dimension is set to 64, with 128 as the dimension of hidden channels. The model uses 2 layers of GINEConv (Brossard et al., 2021). The learning rate for the optimizer was set to 0.0001, with a dropout rate of 0.5 to mitigate overfitting. Training was carried out for 400 epochs. Additionally, STAGE is deployed with 2 layers of GNN on *STAGE-edge-graph* with GINEConv and an edge representation of dimension 32.

## D  AGE REGRESSION EXPERIMENT RESULTS

Table 5 shows that the zero-shot regression on age across different social networks is a challenging task, particularly when the age distributions of the datasets are drastically different. Figure 5 shows that the age distribution in the Pokec dataset is skewed towards younger users, with notable frequencies for ages such as 0 (invalid data), 18, and 20, while ages above 42 are scarcely represented. In contrast, the Friendster dataset contains a much broader range of ages, including significant numbers of users aged in their mid-twenties, such as 25, with smaller frequencies for users up to age 91. This disparity in distribution—where Pokec's frequencies are centered around younger users and Friendster's are

Table 5: Zero-shot test Mean-Square Loss (lower is better) of STAGE and baselines on the soc-pokec dataset with regression tasks on predicting the user's age. Models were trained on the same sample of the Friendster dataset in Section 4. All models show the same bad performance on doing this very challenging task because the root mean squared error (RMSE) of constantly predicting the mean of all age values is 10.7. We use the same configurations as Table 2.

| Model | RMSE ($\downarrow$) $\pm$ Std |
|---|---|
| GINE-structural | 10.99$\pm$0.000 |
| GINE-gaussian | 10.99$\pm$0.000 |
| GINE-normalized | 10.99$\pm$0.000 |
| GINE-llm | 10.99$\pm$0.000 |
| GINE-age | 10.99$\pm$0.000 |
| **GINE-STAGE (Ours)** | **10.99$\pm$0.000** |

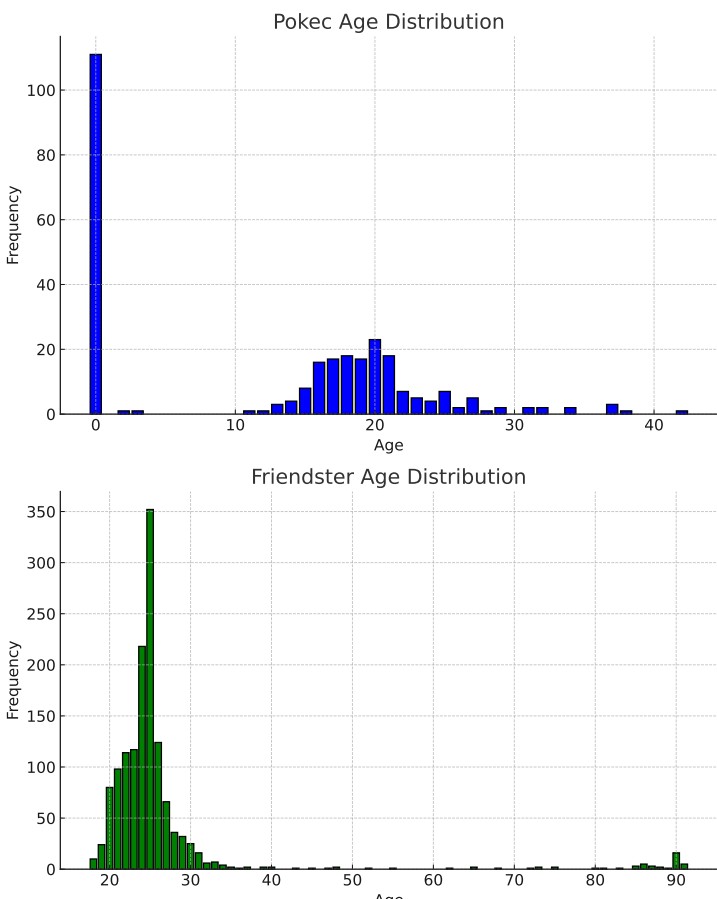

Figure 5: Comparison of Age Distributions in Pokec and Friendster Datasets. The top histogram shows the age distribution for the Pokec dataset, where a significant number of users have an age of 0, followed by a noticeable peak around the age of 20. The bottom histogram illustrates the age distribution for the Friendster dataset, with a strong concentration of users around the age of 25, and a smaller presence of older individuals.

more spread across the adult age spectrum—poses a substantial difficulty for models attempting to generalize across the two networks.

Table 6: Zero-shot test accuracy (higher is better) of STAGE and baselines on the soc-pokec dataset. Models were trained on a sample of the Friendster dataset. **GCN-STAGE demonstrates the best zero-shot test accuracy, surpassing all other methods in both average accuracy and stability.**

| Model | Accuracy ($\uparrow$) $\pm$ **std** |
|---|---|
| GCN-structural | 0.547$\pm$0.0658 |
| GCN-gaussian | 0.567$\pm$0.0382 |
| GCN-normalized | 0.570$\pm$0.0315 |
| GCN-llm | 0.526$\pm$0.0300 |
| GraphAny | 0.591$\pm$0.0083 |
| **GCN-STAGE (Ours)** | **0.593$\pm$0.0046** |

# E    ABLATION STUDY

In this section, we provide ablation studies to further investigate the effectiveness and versatility of STAGE. Experiments in Appendix E.1 complement the main results in the paper by exploring whether STAGE is effective on alternative GNN backbones and configurations. Experiments in Appendix E.2 then study if STAGE can outperform a model trained on the common features shared between train and test domain, validating whether STAGE truly leverages dependencies among unseen features at test time to make predictions.

## E.1    EVALUATING STAGE WITH GCN AS THE BACKBONE GNN

In the main experiments, we employed GINE + NBFNet for link prediction and GINE + GINE for node classification as the backbone GNN configurations. A natural question arises: Can STAGE be effective when using other backbone GNN architectures? To address this, we propose GCN-STAGE (GINE + GCN (Kipf & Welling, 2016)), where we replace the second GINE with a modified GCN to perform message passing on the original graph. We choose GCN as it is a well-known baseline for node classification tasks. We modified GCN to process edge attributes by applying an MLP layer to edge attributes before incorporating them into the edge messages. The first GINE model operating on STAGE-edge-graphs remained unchanged.

Table 6 presents the results, which demonstrate that GCN-STAGE outperforms all baseline methods in terms of average accuracy. Comparing to the other GCN-backbone models, GCN-STAGE outperforms with a 7.33% relative improvement, and achieves an order-of-magnitude smaller standard deviation, showcasing the stability and consistency of predictions across random seeds. Furthermore, same as GINE-STAGE, GCN-STAGE also outperforms GraphAny (Zhao et al., 2024), demonstrating that STAGE is effective on both GCN and GINE. We note that, however, the gain observed with GCN-STAGE is slightly lower than that of GINE-STAGE as shown in Table 2. This is not surprising, as GCN has been shown to have lesser expressivity than GINE (Xu et al., 2018).

These results demonstrate the effectiveness of STAGE regardless of the backbone GNN architecture (GINE or GCN), reinforcing the versatility and general applicability of STAGE across tasks and architectures, further solidifying its strength as a robust framework.

## E.2    COMPARISON WITH MODELS TRAINED ON COMMON FEATURES

In the second ablation study, we aim to investigate whether STAGE is truly leveraging dependencies among multiple unseen node attributes to make zero-shot predictions on the test domain, rather than simply relying on the common attributes shared between train and test. In particular, the attribute "price" and "brand" are shared between the E-commerce datasets (Appendix B.1), and the attribute "age" is shared between Friendster and Pokec (Appendix B.3). Hence, we compare STAGE to a model with the same backbone GNN trained to utilize the shared feature to make predictions. We name these models NBFNet-price on E-commerce datasets for link prediction, and GINE-age on Friendster and Pokec for node classification. We do not experiment with training on the "brand" attribute because its values are distinct (or the distribution have different supports) in different product categories.

Tables 7 and 8 shows the results of this ablation study. NBFNet-STAGE outperforms NBFNet-price with a relative improvement of 69.8% and GINE-STAGE outperforms GINE-age with a relative

Table 7: Zero-shot Hits@1 and MRR of NBFNet-STAGE and NBFNet-price on the E-Commerce dataset. Models are trained on all combinations of four graph domains and tested on the remaining domain. **NBFNet-STAGE significantly outperforms NBFNet-price, demonstrating that STAGE effectively utilizes more information than common feature (price) shared between attribute domains.**

| Model | Hits@1 ($\uparrow$) | MRR ($\uparrow$) |
|---|---|---|
| NBFNet-price | $0.2713 \pm 0.0280$ | $0.3263 \pm 0.0301$ |
| **NBFNet-STAGE (Ours)** | $\mathbf{0.4606 \pm 0.0123}$ | $\mathbf{0.4971 \pm 0.0073}$ |

Table 8: Zero-shot test accuracy of GINE-STAGE and GINE-age on the social network datasets. Models are trained on Friendster and zero-shot tested on Pokec. **GINE-STAGE outperforms GINE-age, demonstrating that STAGE effectively utilizes more information than common feature (age) shared between attribute domains.**

| Model | Accuracy ($\uparrow$) |
|---|---|
| GINE-price | $0.582 \pm 0.0657$ |
| **GINE-STAGE (Ours)** | $\mathbf{0.652 \pm 0.0042}$ |

improvement of 12.0%. These results corroborates our statement that STAGE is capable of leveraging complex dependencies among multiple attributes to make predictions, even when said attributes are unseen during training, as STAGE significantly outperforms the models relying only on shared attributes.

# F  COMPLEXITY ANALYSIS AND RUNTIME COMPARISON

## F.1  STAGE COMPLEXITY

Here we analyze the runtime complexity of STAGE. In particular, we analyze NBFNet-STAGE (used for link prediction) and GINE-STAGE (used for node classification).

Let $p$ be the number of features, $d$ the dimension of internal node and edge embeddings, $|E|$ the number of edges, and $|V|$ the number of nodes in the input graph. For all tasks, STAGE consists of three steps:

1. **Fully Connected STAGE-edge-graph Construction:** This step requires $O(|E|p^2)$ operations because each fully connected STAGE-edge-graph has 2p nodes, and each edge in the original graph induces a fully connected STAGE-edge-graph.

2. **Inference on STAGE-edge-graphs:** We use 2 shared layers of GINE for all STAGE-edge-graphs. A single layer on one fully connected STAGE-edge-graph has complexity $O(pd + p^2d) = O(p^2d)$, since we have $2p$ $d$-dimensional nodes and $(2p)^2$ $d$-dimensional edges in each STAGE-edge-graph. Obtaining edge embeddings across all STAGE-edge-graphs takes $O(|E|p^2d)$.

3. **Inference on the original graph:** For link prediction tasks, we use NBFNet to perform message passing on the original graph, which requires $O(|E|d + |V|d^2)$ for one forward pass (Zhu et al., 2021c). For node classification tasks, we use GINE again, which requires $O(|E|d)$ time.

Hence, in total, running one forward pass has a complexity of $O(|E|p^2d + |E|d + |V|d^2)$ for NBFNet-STAGE, and $O(|E|p^2d + |E|d)$ for GINE-STAGE.

## F.2  TRAINING WALL TIME COMPARISON

The analysis above shows the theoretical complexity of STAGE. Now we study whether STAGE poses a significant computational overhead when deployed in practice. To this end, we measured the average wall time per training epoch of NBFNet-STAGE on the E-Commerce Stores dataset (see Appendix B.1) using an 80GB A100 GPU. We chose this dataset because it is the largest one in our experiment with in total 17463 nodes, 183618 edges, and up to 16 node attributes. Thus it produces the most number of STAGE-edge-graphs and showcases the most explicit runtime contrast. For quick

Table 9: Average per-epoch training time and zero-shot Hits@1 performance of NBFNet-STAGE and baselines on E-Commerce dataset. Time is measured on an 80GB A100 GPU and averaged across 3 training epochs. **NBFNet-STAGE is 7.83% slower than the fastest baseline NBFNet-llm, a reasonable trade-off for its performance gains**.

| Model | Wall Time per Training Epoch (seconds) | Zero-shot Hits@1 on H&M |
|---|---|---|
| NBFNet-raw | 318.65 | $0.0005 \pm 0.0004$ |
| NBFNet-gaussian | 322.13 | $0.0925 \pm 0.0708$ |
| NBFNet-structural | 322.31 | $0.2231 \pm 0.0060$ |
| NBFNet-llm | 316.55 | $0.2302 \pm 0.0015$ |
| NBFNet-normalized | 316.87 | $0.2286 \pm 0.0010$ |
| **NBFNet-STAGE (Ours)** | **341.36** | **$0.4666 \pm 0.0020$** |

comparison, we also display the zero-shot Hits@1 performance when these models are tested on the H&M dataset, which is the result shown in Table 1.

Table 9 displays the runtime comparison results. We observe that NBFNet-STAGE is 7.83% slower than the fastest baseline (NBFNet-llm). Nevertheless, this computational overhead is not extreme but a reasonable tradeoff for its performance gains. The additional time is due to computing STAGE-edge-graph embeddings during each forward pass, while building the STAGE-edge-graphs is a one-time pre-processing step. In practice, the additional factor in the complexity has never prevented us from running in the datasets we considered.

## G ADDITIONAL DISCUSSION OF RELATED WORK

**Graphs Generalization under Distribution Shifts.** Several works address distribution shifts between train and test sets over the same feature space. For instance, (You et al., 2022; Zhu et al., 2021b) employ learned augmentations to mitigate differences in node attribute distributions, training a feature extractor that cannot be used to identify the domain of the node, but these methods are not designed to generalize across distinct attribute spaces. Meanwhile, extensive research has focused on domain adaptation and transfer learning for GNNs (Dai et al., 2022; Li et al., 2020; Kong et al., 2022; Pei et al., 2020; Veličković et al., 2019; Wiles et al., 2022; Zhang et al., 2019; Zhu et al., 2021a), typically assuming access to both source and target domains. In contrast, our work tackles the more challenging scenario of generalizing not only to unseen domains, but also to entirely new attribute spaces.

**Foundation Models for Graph Data.** Foundation models for graph data aim to create versatile graph models capable of generalizing across different graphs and tasks. Despite growing interest, achieving a truly universal graph foundation model remains challenging, especially due to the complexities in designing a suitable graph vocabulary that ensures transferability across datasets and tasks (Mao et al., 2024). Initial efforts in this direction convert attributed graphs into texts and apply an LLM, but this methodology, while promising, risks information loss and may limit transferability (Collins et al., 2024; Gruver et al., 2024; Schwartz et al., 2024). For instance, OFA (Liu et al., 2024) uses frozen LLMs to generate features, and then trains a GNN to perform multiple tasks, while Chen et al. (2024b) explores the potential of LLMs as predictors or enhancersof graph-based predictions. Other methods, like LLaGA (Chen et al., 2024a) and GraphGPT (Tang et al., 2024), use instruction tuning to map graph data into the LLM embedding space. Similarly, Graphtext (Zhao et al., 2023) and Unigraph (He & Hooi, 2024) adopt NLP techniques, with Graphtext (Zhao et al., 2023) translating graphs into natural language via a syntax tree encapsulating node attributes and inter-node relationships, and Unigraph (He & Hooi, 2024) learning a unified graph tokenizer in a self-supervised fashion to generalize across different attribute domains. Prodigy (Huang et al., 2023) further encodes textual features with an LLM and leverages prompt-based graph representations for task generalization.

In contrast, recent approaches forgo LLMs entirely. For instance, Xia & Huang (2024) employs SVD decomposition of the feature matrix to handle shifts to new datasets. Lachi et al. (2024) employs a Perceiver-based encoder to compress domain-specific features into a shared latent space. Zhao et al. (2024) proposes GraphAny, specifically designed for node classification, which models inference

on new graphs as an analytical solution to LinearGNNs, and addresses generalization by learning attention scores to fuse predictions from multiple LinearGNNs. Mao et al. (2023) introduces the concept of feature proximity as a key factor in determining the likelihood of links forming between nodes. Unfortunately, the definition of proximity still depends on the feature space.

Another line of work addresses zero-shot domain transferability on heterogeneous graphs such as knowledge graphs, where both the nodes (entities) and edge types (relations) may be new and unseen on the test-time graph. For instance, ISDEA+ (Gao et al., 2023) proposes a set aggregation layer over the set of edge-type-specific graph representations to ensure equivariance to edge type permutations. Gao et al. (2023) also proposes a theoretical framework named double equivariance that underlies the necessary design principles of models capable of tackling such a task. In contrast, the theoretical framework of our work addresses transferability to unseen attribute domains and proposes a novel connection between statistical tests and the graph regression task. ULTRA (Galkin et al., 2024), on the other hand, builds a relation graph that captures the interactions among different edge types, and applies a two-stage pipeline based on NBFNet (Zhu et al., 2021c) to ensure equivariance to edge type permutations. Similarly, InGram (Lee et al., 2023) also builds a relation graph, but its relation graph differs from ULTRA's in that it computes a set of affinity scores between pairs of relations and use them as edge weights on the relation graph. In comparison, the STAGE-edge-graphs built by our method captures the statistical dependencies among different feature dimensions of node attributes in the graph. However, all of these methods soly rely on graph structure and disregard attributes or features in nodes or edges. In contrast, our work focus on attributed graphs, which is capable of leveraging important information carried in the node attributes.

However, a definitive solution for a universal graph foundation models is yet to arrive, and the search for such a model remains an open challenge.

**Maximal Invariants and Statistical Testing.** Bell (1964) first explored the relationship between invariant and almost-invariant tests in hypothesis testing. Berk & Bickel (1968) and Berk (1970) extended Bell's approach to show that almost-invariant tests are equivalent to invariant ones under certain conditions, which are conditions met in our work. Later, Berk et al. (1996) explored the interplay between sufficiency and invariance in hypothesis testing by providing counterexamples that demonstrate how these concepts, typically assumed to be equivalent under certain conditions, can differ significantly in other scenarios. More recently, Koning & Hemerik (2024) improves the efficiency of hypothesis testing under invariances for large transformation groups such as rotation or sign-flipping without resorting to sampling.

