# OpenReview forum: "Zero-Shot Generalization of GNNs over Distinct Attribute Domains"
_ICLR.cc/2025/Conference — Submitted to ICLR 2025_

### Official Review · Reviewer_iea1 · 2024-11-02

**Soundness:** 3
**Presentation:** 3
**Contribution:** 3
**Rating:** 6
**Confidence:** 3

**Summary:**

The paper introduces STAGE, a novel approach designed to enable zero-shot generalization for Graph Neural Networks (GNNs) across graphs with varying node attribute domains. STAGE aims to learn representation of statistical dependencies between attributes rather than their absolute values. This allows the model to transfer knowledge to unseen domains by leveraging analogous dependencies. Through experiments on multiple datasets, the paper demonstrates STAGE's superior performance in link prediction and node classification tasks, especially in terms of zero-shot cross-domain generalization.

**Strengths:**

- Generalization from a node attributes view and the two stages for processing graph representation are interesting.
- The paper provides a theoretical analysis, linking STAGE with maximal invariants and statistical dependency measures, which provides theoretical support for the model's generalization capabilities.
- The paper shows STAGE's robustness when facing different attribute domains, which is a very important characteristic in the varied real-world data

**Weaknesses:**

- Although the paper presents some quantitative results and shows good performance in different link prediction and node classification domains, it lacks some qualitative analysis. For example, it could demonstrate how the model learns that "income level is positively correlated to phone price" from the training set and then discovers that "height is positively correlated with clothing size" in a new domain, thus generalizing to the new domain.
- STAGE is capable of capturing and leveraging feature dependencies in graph data, rather than relying on specific attribute values. The article can illustrate which feature dependencies are effective on the test set after pre-training the model.

**Questions:**

How well does this model handle larger graph datasets.

---

> ### Author Response · Authors · 2024-11-21
> **Response by Authors (1/2)**
>
> We appreciate the reviewer's positive assessment of our work, particularly their acknowledgement of its theoretical support, and its relevance to real-world applications. We have carefully considered the reviewer's questions and provide our responses below:
>
> > W1/W2: Qualitative Analysis on Feature Dependencies
>
> Thank you for the valuable suggestion to illustrate how STAGE learns and generalizes feature dependencies. We conducted a feature-isolation experiment on the node classification task using the Friendster and Pokec datasets. Our results demonstrate STAGE's ability to learn and prioritize relevant features to perform prediction tasks.
>
> Below we summarize our findings, which we will add to the Appendix in our revision.
>
> **Experiment Setup**
>
> In this feature-isolation experiment, we systematically removed individual attributes (e.g., age, interest, occupation, music, tv) from the Friendster training dataset and trained GINE-STAGE on these reduced datasets.. Zero-shot performance was then evaluated on the Pokec dataset for predicting the "gender" label. The following table displays the results:
>
> | Individual feature removed from Friendster    | age   | interest | occupation | music | tv    | (None removed) |
> |-----------------------------------------------|-------|----------|------------|-------|-------|----------------|
> | Test accuracy of predicting "gender" on Pokec | 0.500 | 0.632    | 0.632      | 0.623 | 0.649 | 0.649          |
>
> **Findings**
> Removing certain features (e.g., “tv”) during training has minimal impact on test performance (predicting “gender”), indicating that STAGE is able to learn during training that certain features are not important for the task.

---

> ### Author Response · Authors · 2024-11-21
> **Response by Authors (2/2)**
>
> > Q1: How well does STAGE handle larger graph datasets
>
> We thank the reviewer for raising this important point about STAGE's scalability to larger graph datasets. We have conducted a thorough analysis and gathered empirical evidence to address this concern. These findings, along with a detailed discussion of our approach to handling large-scale graphs, are included in the updated manuscript (which will be available during the rebuttal period).
>
> **Complexity Analysis**
>
> Let  $p$ be the number of features, $d$ the dimension of internal node and edge embeddings, $|E|$ the number of edges, and $|V|$ the number of nodes in the input graph. For the link prediction task, STAGE consists of three steps:
>
> 1. **Fully Connected STAGE-Edge-Graph Construction:** This step requires  $O(|E| p^2)$ operations because each fully connected STAGE-edge-graph has 2p nodes.
> 2. **Inference (GINE Layers):** We use 2 shared layers of GINE [1] for all STAGE-edge-graphs. A single layer on one fully connected STAGE-edge-graph has complexity $O(p d + p^2 d) = O(p^2 d)$ since we have 2p d-dimensional nodes and $(2p)^2$ d-dimensional edges in each graph. Aggregating edge embeddings across all graphs takes $O(|E| p^2 d)$.
> 3. **Inference (NBFNet):** We use NBFNet to perform message passing on the original graph, which requires $O(|E|d + |V|d^2)$ for one forward pass [2].
>
> **Total Complexity:** The overall forward pass has a complexity of $O(|E| p^2 d + |E|d + |V|d^2)$.
>
> **Runtime Comparison**
> We measured the average wall time per training epoch on the E-Commerce Stores dataset, the largest dataset among our experiments. It contains a total of 17463 nodes, 183618 edges, and up to 16 node attributes. We run all models on the 80GB NVIDIA A100 GPU. The following table summarizes the runtime of each model and the their respective zero-shot test performance on the H&M dataset for reference.
>
> | Models          	| Wall Time per Training Epoch on E-Commerce (seconds) | Zero-shot Hits@1 Performance on H&M |
> |---------------------|----------------------------------------|----------------------------------------|
> | NBFNet-raw      	| 318.65 | 0.0005 $\pm$ 0.0004 |
> | NBFNet-gaussian 	| 322.13 | 0.0925 $\pm$ 0.0708 |
> | NBFNet-structural   | 322.31 |  0.2231 $\pm$ 0.0060 |
> | NBFNet-llm      	| 316.55 | 0.2302 $\pm$ 0.0015 |
> | NBFNet-normalized   | 316.87 | 0.2286 $\pm$ 0.0010 |
> | **NBFNet-STAGE (Ours)** | **341.36** | **0.4666 $\pm$ 0.0020** |
>
> NBFNet-STAGE is only 7.83% slower than the fastest baseline (NBFNet-llm), a reasonable tradeoff for its performance gains. The additional time is due to computing STAGE-edge-graph embeddings during each forward pass while building the STAGE-edge-graphs is a one-time preprocessing step. In practice, the additional factor in the complexity has never prevented us from running in the datasets we considered.
>
> In summary, to answer the reviewer’s question, we expect that STAGE can scale quite well to graphs with a large number of nodes (to tens of thousands and more) with a moderate number of features (smaller than 100). On the other hand, scalability is more limited on graphs with thousands of features. Nevertheless, one can use techniques to mitigate this issue, including but not limited to feature selection by performing an association study or reducing attribute dimensions via techniques such as Principal Component Analysis (PCA). We have added this discussion in the Limitation & Future Work section of the updated paper, which we will upload during the rebuttal period.
>
>
> [1] Zhu et al. "Neural bellman-ford networks: A general graph neural network framework for link prediction." NeurIPS 2021.

---

> > ### Comment · Reviewer_iea1 · 2024-11-25
> >
> > Thanks for your response! Good work has been made about Complexity Analysis and Runtime Comparison. My concern has been resolved. I will keep my positive rating.

---

### Official Review · Reviewer_U75f · 2024-11-02

**Soundness:** 2
**Presentation:** 3
**Contribution:** 3
**Rating:** 5
**Confidence:** 4

**Summary:**

The paper presents STAGE, a method that enables zero-shot generalization of graph neural networks (GNNs) across graphs with different attribute domains. STAGE constructs STAGE-edge-graphs to capture statistical dependencies between attributes instead of absolute values, facilitating transferability to unseen domains. The method shows substantial improvement in zero-shot tasks like link prediction and node classification on graphs with entirely new feature spaces.

**Strengths:**

1. STAGE's strategy to use statistical dependencies rather than raw attribute values to enhance zero-shot generalization in GNNs is novel for graph machine learning.
2. The theoretical basis for STAGE, connecting maximal invariants and statistical dependencies, is well-articulated and provides a sound foundation for the empirical results.
3. STAGE is shown to be adaptable across domains of varied attribute types and dimensions, a crucial quality for real-world applicability.

**Weaknesses:**

1. STAGE's two-stage process involving STAGE-edge-graphs, conditional probability matrices, and subsequent embeddings may be challenging to implement or optimize in practice. Details on computational overhead compared to baselines would enhance clarity.
2. While STAGE is effective for pairwise dependencies, it is unclear how it handles more complex dependencies in highly interconnected graphs.
3. The success of STAGE appears dependent on the architecture and expressivity of the underlying GNNs (M1 and M2). Sensitivity analysis on different GNN backbones might clarify robustness across architectures.
4. The evaluation focuses on e-commerce and social network datasets; examining STAGE’s generalizability on domains like biomedical or geospatial networks would strengthen claims of universality.

**Questions:**

1. How does STAGE handle attribute domains with highly heterogeneous data types, such as unstructured or mixed media data?
2. Could the authors elaborate on the computational cost associated with STAGE compared to baselines, especially in large-scale graphs?
3. Does STAGE's reliance on GNN backbones like NBFNet affect its generalizability? Could alternative GNN architectures be equally effective?
4. Has STAGE been evaluated in terms of the interpretability of the learned dependencies? If so, what methods were used?

---

> ### Author Response · Authors · 2024-11-21
> **Response by Authors (1/2)**
>
> We would like to thank the reviewer for their appreciation that this work is addressing a novel problem, has well-articulated theoretical support, and the proposed method is crucial for real-world applicability. We will now address your questions as follows:
>
> > W1/Q2: Computational Overhead
>
> We appreciate this critical observation. In response to your request, we have investigated the training wall time of STAGE compared to the baseline models.
> We measured the average wall time per training epoch on the E-Commerce Stores dataset used in the experiments shown in the following table, using an 80GB A100 GPU. This dataset is the largest in our experiment with a total of 17463 nodes, 183618 edges, and up to 16 node attributes. Thus it produces the most number of STAGE-edge-graphs and showcases the most explicit runtime contrast. The times are reported in the table below.
> | Models          	| Wall Time per Training Epoch (seconds) | Hits@1 |
> |---------------------|----------------------------------------|------------|
> | NBFNet-raw      	| 318.65                             	| 0.0000 $\pm$ 0.0000 |
> | NBFNet-gaussian 	| 322.13                        	| 0.2101 $\pm$ 0.0428 |
> | NBFNet-structural   | 322.31                             	| 0.3149 $\pm$ 0.0253 |
> | NBFNet-llm      	| 316.55                             	| 0.3226 $\pm$ 0.0190 |
> | NBFNet-normalized   | 316.87                             	| 0.3269 $\pm$ 0.0213 |
> | **NBFNet-STAGE (Ours)** | **341.36**                             	| **0.4606 $\pm$ 0.0123** |
>
> NBFNet-STAGE is 7.83% slower than the fastest baseline (NBFNet-llm), a reasonable tradeoff for its performance gains. The additional time is due to computing STAGE-edge-graph embeddings during each forward pass, while building the STAGE-edge-graphs is a one-time preprocessing step. In practice, the additional factor in the complexity has never prevented us from running in the datasets we considered.
>
> > W2: How STAGE handles complex dependencies in highly interconnected graphs
>
> Thank you for the opportunity to clarify this point.
>
> We would like to emphasize that the STAGE-edge-graph construction is theoretically capable of representing complex and high-order dependencies between multiple node attributes. In section 3.1, we motivate the theory by discussing the two-sample independence tests as an example, but the theory **applies to more complex interactions, such as higher-order conditional independence tests**. This is because many statistical tests for independence and conditional independence are all equivalent to rank tests, which disregard specific feature values and focus solely on their relative rankings [1]. Theorem 3.3 then formally demonstrates the expressiveness of STAGE-edge-graph by proving that applying a maximally expressive GNN encoder to each STAGE-edge-graph (along with positional encodings) and subsequently using a most expressive multiset encoder on the resulting embeddings allows for the approximation of *any* statistical test measuring dependencies or interactions between any number of node attributes.
>
> We appreciate this comment and, in the updated paper, we emphasize that the theory applies not only to two-sample tests but also to any high-order statistical tests measuring complex dependencies. We will upload the updated paper during the rebuttal period.
>
> > W3/Q3: Sensitivity Analysis of Different GNN Backbones
>
> This is indeed a very good point worthy of studying. Following your suggestion,  we have experimented with other GNN backbones on the node classification experiments we reported in the main paper. These analysis are discussed in updated **Appendix E.1**, which we summarize below.
>
> To demonstrate STAGE's adaptability, we replaced the GINE model used for the node classification experiment with a Graph Convolutional Network (GCN) adapted to handle multi-dimensional edge attributes. Specifically, an MLP was integrated into GCN’s message-passing to process edge attributes. The results, shown in updated Table 6 (and the attached table below), confirm that GCN-STAGE significantly outperforms all baselines, achieving a 7.33% improvement in average zero-shot test accuracy and a much smaller standard deviation, showcasing its robustness and stability.
>
> | Model            | Accuracy ($\uparrow$) |
> |------------------|----------------------------|
> | GCN-structural   | 0.547 $\pm$ 0.0658              |
> | GCN-gaussian     | 0.567 $\pm$ 0.0382              |
> | GCN-normalized   | 0.570 $\pm$ 0.0315              |
> | GCN-llm          | 0.526 $\pm$ 0.0300              |
> | **GCN-STAGE (Ours)** | **0.593** $\pm$ **0.0046**          |
>
> These results demonstrate that STAGE improves performance over all baseline methods, regardless of the backbone GNN architecture (GINE or GCN). This reinforces the versatility and general applicability of STAGE across tasks and architectures, further solidifying its strength as a robust framework.

---

> ### Author Response · Authors · 2024-11-21
> **Response by Authors (2/2)**
>
> > W4: STAGE’s Generalizability on Biomedical or Geospatial Domains
>
> Thank you for this suggestion. We believe that studying the generalizability of STAGE to these domains represents an interesting and important aspect, which we are eager to conduct as a future research. We have incorporated this suggestion in the updated Limitations & Future Work section.
>
>
> > W5: Handling Highly Heterogeneous Data
>
> A5: STAGE's efficacy is built upon the theory of maximal invariants, which is currently limited to scalar and discrete variables. Extending this theory to handle unstructured or mixed media data is an exciting but challenging topic, which we leave as future work. Our current approach focuses on attributes with values in one-dimensional real space, though nodes may have multiple attributes. Addressing multimedia data embeddings would require substantial methodological modifications.
>
> > W6: Interpreting STAGE’s Learned Dependencies
>
> A6: We appreciate your insightful suggestion regarding the interpretability of STAGE's learned dependencies. To address this, we conducted a feature-isolation experiment on the node classification task using the Friendster and Pokec datasets. Our results demonstrate STAGE's ability to learn and prioritize relevant features to perform prediction tasks.
>
> Below we summarize our findings, which we will add to the Appendix in our revision.
>
> **Experiment Setup**
>
> In this feature-isolation experiment, we systematically removed individual attributes (e.g., age, interest, occupation, music, tv) from the Friendster training dataset and trained GINE-STAGE on these reduced datasets.. Zero-shot performance was then evaluated on the Pokec dataset for predicting the "gender" label. The following table displays the results:
>
> | Individual feature removed from Friendster    | age   | interest | occupation | music | tv    | (None removed) |
> |-----------------------------------------------|-------|----------|------------|-------|-------|----------------|
> | Test accuracy of predicting "gender" on Pokec | 0.500 | 0.632    | 0.632      | 0.623 | 0.649 | 0.649          |
>
> **Findings**
> Removing certain features (e.g., “tv”) during training has minimal impact on test performance (predicting “gender”), indicating that STAGE is able to learn during training that certain features are not important for the task.
>
>
>
> We thank the reviewer for their thoughtful comments, and we believe these revisions significantly strengthen our paper. Thank you for your time and support!
>
>
> [1] Bell. "A characterization of multisample distribution-free statistics." AMS 1964.
>
> [2] Hu et al. "Strategies for pre-training graph neural networks." ICLR 2020.
>
> [3] Zhu et al. "Neural bellman-ford networks: A general graph neural network framework for link prediction." NeurIPS 2021.

---

> > ### Comment · Reviewer_U75f · 2024-11-27
> >
> > Thanks for your response.
> >
> > However, part of my concerns are not addressed. Specifically, the E-Commerce Stores dataset is not large enough to demonstrate the computational overhead. The issues of W4 and my questions have not been solved.
> >
> > Therefore, I will keep my rating.

---

> ### Author Response · Authors · 2024-11-30
> **Response by Authors (3/3)**
>
> Dear Reviewer,
>
> Thank you for your thoughtful follow-up comments and continued engagement with our work on STAGE. We are glad that you recognize the novelty and theoretical foundation of our approach and appreciate the constructive feedback.
>
> ### W4: Applying STAGE to Geospatial and Biomedical datasets
>
> We would be happy to explore applying STAGE to Geospatial and Biomedical datasets if the reviewer could point us to two such datasets with rich and distinct feature spaces. We note, for instance, that some datasets, such as AirBrazil, AirEU, and AirUS [1], have no node features (they use one-hot encodings of node ids as features, which are provably not transferable zero-shot).
>
> ### Scalability Analysis
>
> We would like to expand on our analysis of the computational complexity of STAGE for the link prediction task. Let's denote the number of features as $p$, the dimension of internal node and edge embeddings as $d$, the number of edges as $|E|$, and the number of nodes as $|V|$.
>
> #### Forward Pass Complexity
>
> The forward pass of STAGE consists of three main steps:
>
> 1. **Fully Connected STAGE-Edge-Graph Construction**: This step requires $O(|E| p^2)$ operations, as each fully connected STAGE-edge-graph has 2p nodes.
> 2. **Inference (GINE Layers)**: We use two shared layers of GINE for all STAGE-edge-graphs. The complexity of a single layer on one fully connected STAGE-edge-graph is $O(p^2 d)$, since we have 2p d-dimensional nodes and $(2p)^2$ d-dimensional edges in each graph. Aggregating edge embeddings across all graphs takes $O(|E| p^2 d)$.
> 3. **Inference (NBFNet)**: We use NBFNet to perform message passing on the original graph, which requires $O(|E|d + |V|d^2)$ for one forward pass.
>
> #### Total Complexity
>
> **Total Complexity**: The overall forward pass has a complexity of $O(|E| p^2 d + |E|d + |V|d^2)$. As we can see, the complexity is linear in the number of nodes and edges (graph size), indicating that graph size is less of an issue than the number of features.
>
> Thank you again for your feedback, and we look forward to further discussions.
>
> [1] Ribeiro et al. "struc2vec: Learning Node Representations from Structural Identity." Proceedings of the 23rd ACM SIGKDD International Conference on Knowledge Discovery and Data Mining, ACM, 2017.

---

### Official Review · Reviewer_JvnB · 2024-11-03

**Soundness:** 3
**Presentation:** 2
**Contribution:** 2
**Rating:** 5
**Confidence:** 3

**Summary:**

This paper introduces STAGE, a method designed for zero-shot generalization across attributed graphs with distinct attribute domains. STAGE constructs what it calls STAGE-edge-graphs for each edge in a graph, embedding statistical dependencies between attributes at each node pair. The model achieves significant performance gains in zero-shot settings for tasks like link prediction and node classification on various datasets.

**Strengths:**

1. This paper achieves SOTA results by embedding statistical dependencies rather than raw features.
2. The STAGE is a domain-agnostic framework, which can generalize across disparate attribute spaces.

**Weaknesses:**

1. The STAGE-edge-graph is a fully connected weighted graph, so I am concerned about the complexity.
2. Edge-based embeddings may limit its ability to capture high-order interactions in graphs.
3. The motivation in the introduction is not presented well. The authors didn't analyze why their proposed method can address the limitations they mentioned before, so it's hard to understand the intrinsic research thinking.
4. The experiments are a little weak. For example, I believe the 4.2 and 4.3 belong to the same type of experiment, they didn't analyze the complexity and the ablation study, and they didn't include the limited research papers they mentioned in the introduction into baselines,  which weakens the convincing.
5. I noticed this paper was submitted to ICML workshop so there is authors' information leakage. Both papers present STAGE for zero-shot generalization of GNNs across different attribute domains, and this paper just extends some real-world testing datasets.
6. The authors didn't release their code although this is not compulsory, which may limit their reproducibility.

**Questions:**

Please see the weaknesses

---

> ### Author Response · Authors · 2024-11-21
> **Response by Authors (1/4)**
>
> Thank you for your thoughtful review effort and constructive comments. We are pleased to see you appreciated the generality of our framework and recognized the empirical improvements it yields. Nonetheless, the review raised a number of important questions, which we answer in the following.
>
> > W1: The STAGE-edge-graph is fully connected so I am concerned about the complexity.
>
>
> A1: We thank the reviewer for raising this important point. We have conducted a comprehensive complexity analysis and runtime comparison, which will be included in the updated paper (to be uploaded during the rebuttal period). A summary of our findings is provided below.
>
> **Complexity Analysis**
>
> Let  $p$ be the number of features, $d$ the dimension of internal node and edge embeddings, $|E|$ the number of edges, and $|V|$ the number of nodes in the input graph. For the link prediction task, STAGE consists of three steps:
>
> 1. **Fully Connected STAGE-Edge-Graph Construction:** This step requires  $O(|E| p^2)$ operations because each fully connected STAGE-edge-graph has 2p nodes.
> 2. **Inference (GINE Layers):** We use 2 shared layers of GINE [1] for all STAGE-edge-graphs. A single layer on one fully connected STAGE-edge-graph has complexity $O(p d + p^2 d) = O(p^2 d)$ since we have 2p d-dimensional nodes and $(2p)^2$ d-dimensional edges in each graph. Aggregating edge embeddings across all graphs takes $O(|E| p^2 d)$.
> 3. **Inference (NBFNet):** We use NBFNet to perform message passing on the original graph, which requires $O(|E|d + |V|d^2)$ for one forward pass [2].
>
> **Total Complexity:** The overall forward pass has a complexity of $O(|E| p^2 d + |E|d + |V|d^2)$.
>
> **Runtime Comparison**
> We measured the average wall time per training epoch on the E-Commerce Stores dataset used in the experiments shown in the following table, using an 80GB A100 GPU. This dataset is the largest in our experiment with 17463 nodes, 183618 edges, and up to 16 node attributes. Thus it produces the most number of STAGE-edge-graphs and showcases the most explicit runtime contrast.
>
> | Models          	| Wall Time per Training Epoch on E-Commerce (seconds) | Zero-shot Hits@1 Performance on H&M |
> |---------------------|----------------------------------------|----------------------------------------|
> | NBFNet-raw      	| 318.65 | 0.0005 $\pm$ 0.0004 |
> | NBFNet-gaussian 	| 322.13 | 0.0925 $\pm$ 0.0708 |
> | NBFNet-structural   | 322.31 |  0.2231 $\pm$ 0.0060 |
> | NBFNet-llm      	| 316.55 | 0.2302 $\pm$ 0.0015 |
> | NBFNet-normalized   | 316.87 | 0.2286 $\pm$ 0.0010 |
> | **NBFNet-STAGE (Ours)** | **341.36** | **0.4666 $\pm$ 0.0020** |
>
> NBFNet-STAGE is 7.83% slower than the fastest baseline (NBFNet-llm), a reasonable tradeoff for its performance gains. The additional time is due to computing STAGE-edge-graph embeddings during each forward pass, while building the STAGE-edge-graphs is a one-time preprocessing step. In practice, the additional factor in the complexity has never prevented us from running in the datasets we considered.
>
> **Scalability Considerations**
> Despite the potential scalability challenges posed by graphs with thousands of features, we believe STAGE-edge-graph remains a viable solution. In the revised Limitations and Future Work section, we outline feature selection strategies (e.g., association studies) as effective mitigation techniques.
>
> In summary, while STAGE-edge-graph introduces a modest overhead, its scalability is manageable with appropriate feature selection techniques, making it feasible for practical deployment.

---

> ### Author Response · Authors · 2024-11-21
> **Response by Authors (2/4)**
>
> > W2: Edge-based embeddings may limit its ability to capture high-order interactions in graphs.
>
> A2: We appreciate the reviewer’s comment regarding the potential limitations of edge-based embeddings in capturing high-order interactions. To ensure we fully understand your question and address it effectively, could we kindly ask the reviewer to please provide further details or examples of specific high-order interaction patterns that you believe might be challenging for our approach to capture?
>
> While we await the reviewer’s clarification, we can emphasize that the STAGE-edge-graph construction is theoretically capable of representing complex and high-order dependencies between multiple node attributes. Theorem 3.3 formally demonstrates this capability by proving that applying a maximally expressive GNN encoder to each STAGE-edge-graph (along with positional encodings) and subsequently using a most expressive multiset encoder on the resulting embeddings allows for the approximation of any statistical test measuring dependencies or interactions between any number of node attributes.
>
> This theoretical foundation suggests that the STAGE-edge-graph itself is not the limiting factor in terms of expressivity. If a particular implementation utilizing STAGE-edge-graphs struggles to capture high-order interactions, it is more likely due to limitations in the chosen backbone neural network architectures rather than the inherent nature of the edge-based embeddings.
>
> > W3. The motivation in the introduction is not presented well. The authors didn't analyze why their proposed method can address the limitations they mentioned before, so it's hard to understand the intrinsic research thinking.
>
> A3: We appreciate the reviewer's feedback and have revised the introduction to more clearly articulate the motivation behind our approach and its ability to generalize to new domains.
>
> The core innovation of this work lies in addressing the challenge of zero-shot generalization across diverse attribute spaces. To achieve this, we identify three key invariances that representations must possess: invariance to changes in attribute values, permutation of attribute identities, and permutation of node identities. STAGE constructs attribute representations that capture maximal information about dependencies and interactions among raw attribute values while adhering to these invariances.
>
> This challenge is related to the concept of rank tests and maximal invariants from statistics. Notably, many statistical tests for independence and conditional independence are equivalent to rank tests, which disregard specific feature values and focus solely on their relative rankings [3]. Inspired by this insight, we developed the theoretical formulation of the feature hypergraph (Definition 3.1), a graphical representation of rank tests. Subsequently, we introduced STAGE-edge-graphs as a practical implementation that, as demonstrated by Theorem 3.3,  possesses equivalent expressivity to the feature hypergraph while offering computational efficiency over the equivalent rank test (feature hypergraph).
>
> > W4.1 I believe the 4.2 and 4.3 belong to the same type of experiment
>
> A4: The experiments in Sections 4.2 and 4.3 address different tasks: Section 4.2 focuses on link prediction, while Section 4.3 evaluates node classification. Furthermore, they are conducted on distinct datasets (E-commerce and HM for link prediction; Friendster and Pokec for node classification) and use different backbone GNNs within the STAGE framework.
>
> This deliberate separation of tasks, datasets, and GNN backbones demonstrates the adaptability of the STAGE-edge-graph construction strategy across diverse settings. The consistent effectiveness observed across these varied experimental configurations reinforces the robustness and generalizability of STAGE.

---

> ### Author Response · Authors · 2024-11-21
> **Response by Authors (3/4)**
>
> > W4.2 Analysis of ablation study
>
> A5: Thank you for your valuable feedback. In addition to the experiments evaluating alternative featurization strategies (e.g., NBFNet-structural, NBFNet-gaussian, NBFNet-normalized), we have now included two new ablation studies in **Appendix E** to further illustrate STAGE’s flexibility and effectiveness. Below, we summarize the first study:
>
> **Extra Ablation 1: STAGE's Flexibility Across GNN Backbones**
>
> In this ablation we study whether STAGE is still effective on alternative GNN backbone models. We chose the node classification task for this investigation, where we train on the Friendster dataset and zero-shot test on the Soc-Pokec dataset. To demonstrate STAGE's adaptability, we replaced the GINE backbone that was used for our results shown in Table 2 with a Graph Convolutional Network (GCN) adapted to handle multi-dimensional edge attributes. Specifically, an MLP was integrated into GCN’s message-passing to process edge attributes.
>
> The results, shown in updated Table 6, confirm that GCN-STAGE significantly outperforms all baselines, achieving a 7.33% improvement in average zero-shot test accuracy and a much smaller standard deviation, showcasing its robustness and stability.
>
> | Model            | Accuracy ($\uparrow$) |
> |------------------|-----------------------------------|
> | GCN-structural   | 0.547 $\pm$ 0.0658              |
> | GCN-gaussian     | 0.567 $\pm$ 0.0382              |
> | GCN-normalized   | 0.570 $\pm$ 0.0315              |
> | GCN-llm          | 0.526 $\pm$ 0.0300              |
> | **GCN-STAGE (Ours)** | **0.593** $\pm$ **0.0046**          |
>
> These results demonstrate the effectiveness of STAGE regardless of the backbone GNN architecture (GINE or GCN), reinforcing the versatility and general applicability of STAGE across tasks and architectures, further solidifying its strength as a robust framework.
>
> **Extra Ablation 2: Leveraging Shared Features Across Datasets.**
>
> In this second ablation study, we aim to investigate whether STAGE is truly leveraging dependencies among multiple unseen node attributes to make zero-shot predictions, rather than simply relying on the common attributes shared between train and test.
>
> Across E-commerce datasets (excluding H&M), product price serves as a common attribute. We trained NBFNet-STAGE and a baseline model, NBFNet-Price, which utilizes only price information. As detailed in the updated Table 1 (also see the attached table below), NBFNet-STAGE achieves a remarkable 70% improvement over NBFNet-Price in Hits@1. This substantial gain underscores STAGE's ability to extract valuable insights from complex graph relationships that go beyond simple shared features.
>
> **In the original submission we already did this ablation study for the Pokec and Friendster datasets**, where we trained a GINE (GINE -Age) utilizing only age as input and compared its performance to our GINE-STAGE model. As shown in (the original) Table 2 (and in the table below), GINE-STAGE outperforms GNN-Age by a significant margin of 12% in accuracy. This highlights that STAGE effectively leverages the rich relational information within the graph structure, surpassing the predictive power attainable solely from shared node attributes.
>
> For your quick reference, here are the simplified tables comparing NBFNet-Price with NBFNet-STAGE and GINE-Age with GINE-STAGE:
>
> | Models                  | Test Hits@1 on Held-out E-Comm. Store | Test MRR on Held-out E-Comm. Store |
> |-------------------------|---------------------------------------|------------------------------------|
> | NBFNet-price            | 0.2713 $\pm$ 0.0280                   | 0.3263 $\pm$ 0.0301                |
> | **NBFNet-STAGE (Ours)** | **0.4606 $\pm$ 0.0123**               | 0.4971 $\pm$ 0.0073                |
>
> | Models                | Test Accuracy on Pokec |
> |-----------------------|------------------------|
> | GINE-age              | 0.582 $\pm$ 0.0657     |
> | **GINE-STAGE (Ours)** | **0.652 $\pm$ 0.0042** |

---

> ### Author Response · Authors · 2024-11-21
> **Response by Authors (4/4)**
>
> > W4.3. They didn't include the limited research papers they mentioned in the introduction into baselines
>
> We want to first clarify that the NBFNet-llm (shown in Table 1 for link prediction) and the GINE-llm (shown in Table 2 for node classification) is the PRODIGY [4] baseline, which we mentioned as one of the text encoder methods in the introduction.  The other two text encoder methods, OneForAll [5] and LLaGA[6], cannot be easily adapted for a fair comparison against our method and, although feasible, such an effort lies outside of the scope of the present work. For instance, on the link prediction task, OneForAll in its current implementation is only designed to classify edge types (or relation types) given a source and target node, i.e., answering queries of the form (s, ?, t), and the authors reported results using classification accuracy as the metric. However, it does not support the more common task of predicting the tail node given the source node and edge type, i.e., answering queries of the form (s, r, ?), and does not support ranking-based metrics such as Hits@1, MRR, etc., which are the metrics we used in our work. On the other hand, LLaGA relies on LLMs to make final predictions by converting graphs into inputs compatible with LLMs, and the context windows of the LLMs used in their work are not large enough to accommodate the large-scale graphs we have in our dataset, e.g. the E-Commerce Store dataset.
>
> Nonetheless, we welcome the reviewer’s suggestion and included GraphAny [7] as an additional baseline in the updated experiment results. Since GraphAny is specifically designed for the node classification task, we report its results in the zero-shot node classification experiment shown in Table 2 of the updated paper.
>
> The following is the updated Table 2 incorporating GraphAny’s results:
>
> | Models        	| Accuracy ($\uparrow$) |
> |-------------------|---------------------------------|
> | GINE-structural | 0.564 $\pm$ 0.0466 |
> | GINE-gaussian | 0.588 $\pm$ 0.0250 |
> | GINE-normalized | 0.541 $\pm$ 0.0148 |
> | GINE-age | 0.582 $\pm$ 0.0657 |
> | GINE-llm [3] | 0.550 $\pm$ 0.0368 |
> | GraphAny [4]      	| 0.591 $\pm$ 0.0083          	|
> | **GINE-STAGE (Ours)** | **0.652 $\pm$ 0.0042**          	|
>
> We observe that GraphAny is the best-performing model among all the baseline models. Nevertheless, our mode GINE-STAGE significantly outperforms GraphAny with a 10.3% relative performance improvement. This demonstrates the effectiveness and superior performance of STAGE compared to the baseline node classification foundation model.
>
> > W5. I noticed this paper was submitted to an ICML workshop…
>
>
> A6: Thank you for raising this point. We have discussed this matter with the Area Chair, who has confirmed that our submission adheres to ICLR's policies. We hope this is enough to dispel these concerns.
>
> > W6. The authors didn't release their code although this is not compulsory, which may limit their reproducibility.
>
> A7: Thank you for this question. We plan to publicly release our code upon acceptance.
>
> We believe we have carefully addressed all reviewer feedback, providing compelling evidence for the effectiveness and generalizability of our STAGE approach. We thank the reviewer for their thoughtful comments, and we believe these revisions significantly strengthen our paper. Thank you! We hope the reviewer will find these changes satisfactory to reconsider their score. We are more than happy to address any further questions.
>
> [1] Hu et al. "Strategies for pre-training graph neural networks." ICLR 2020.
>
> [2] Zhu et al. "Neural bellman-ford networks: A general graph neural network framework for link prediction." NeurIPS 2021.
>
> [3] Bell. "A characterization of multisample distribution-free statistics." AMS 1964.
>
> [4] Huang et al. "Prodigy: Enabling in-context learning over graphs." NeurIPS 2024.
>
> [5] Liu, et al. “One for all: Towards training one graph model for all classification tasks.” ICLR 2024.
>
> [6] Chen, et al. “LLaGA: Large language and graph assistant.” ICML, 2024.
>
> [7] Zhao et al. "Graphany: A foundation model for node classification on any graph." ArXiv 2024.

---

> ### Comment · Reviewer_JvnB · 2024-11-21
>
> Thanks for the authors' response. My questions are addressed, and I think it's good to include the complexity analysis. I will increase my score.

---

> > ### Author Response · Authors · 2024-11-21
> > **Thank you!**
> >
> > Dear Reviewer,
> >
> > Thank you for your thoughtful feedback and for increasing the score of our manuscript. We appreciate your insightful questions, whose answers have strengthened our paper. The complexity analysis will be included in the revised submission alongside the other results.
> >
> > We welcome any further questions you may have.
> >
> > Sincerely,
> > The Authors

---

### Official Review · Reviewer_nStQ · 2024-11-05

**Soundness:** 3
**Presentation:** 3
**Contribution:** 3
**Rating:** 6
**Confidence:** 4

**Summary:**

This paper studies how to use a pre-trained graph model in any new domain with unseen attributes, enhancing the zero-shot generalization. The authors propose a new model STAGE by learning the representations of statistical dependency between attributes, instead of the attribute values themselves. They also conduct experiments to validate the performance of STAGE across several benchmark datasets.

**Strengths:**

1.	The paper is well written and easy-to-follow.
2.	Extensive results validate the effectiveness of the proposed model.

**Weaknesses:**

1.	More discussions on the variants on LLM should be made.

**Questions:**

N/A

---

> ### Author Response · Authors · 2024-11-21
> **Response by Authors**
>
> We thank the reviewer for appreciating the presentation of our work and the extensive empirical results. We address the question in detail below.
>
> > W1. More discussions on the variants on LLM should be made.
>
> A1: Thank you for your question. Our baseline models, NBFNet-LLM and GINE-LLM, essentially implement PRODIGY [1] but use all-MiniLM-L6-v2 for node feature embedding instead of RoBERTa. We chose this alternative due to its specialized training on sentence embeddings and speed, which offers advantages in capturing semantic relationships compared to RoBERTa (an older model).
>
> Nonetheless, pretrained language models can be integrated in other ways than encoding textified features. For instance, in our introduction we mentioned OneForAll [2] and LLaGA [3]. While OneForAll's current implementation excels at classifying edge types given source and target nodes, it lacks support for predicting tail nodes given source nodes and edge types – a requirement for our tasks and needed to support ranking-based metrics such as Hits@1, MRR. Furthermore, LLaGA's reliance on LLMs with limited context windows poses a significant obstacle when dealing with the large-scale graphs in our dataset, particularly the E-Commerce Store dataset.
>
> Therefore, while we recognize the value of exploring alternative LLM integration strategies, focusing on the NBFNet-LLM and GINE-LLM baselines allows us to grasp the limitations of directly incorporating node feature information through textification within a graph neural network framework.
>
> We appreciate the reviewer for bringing up this point for discussion, and we have added a discussion in the Limitation & Future Work section of the updated paper. We will upload the updated paper during the rebuttal period.
>
>
>
> [1] Qian, et al. “Prodigy: Enabling in-context learning over graphs.” NeurIPS 2024.
>
> [2] Liu, et al. “One for all: Towards training one graph model for all classification tasks.” ICLR 2024.
>
> [3] Chen, et al. “LLaGA: Large language and graph assistant.” ICML, 2024.

---

### Author Response · Authors · 2024-11-21
**General Response and New Experiment Results (2/2)**

> **STAGE’s scalability to large graphs**

As per JvnB and U75f’s request, we offered a complexity analysis and a comparison of training time between STAGE and the baselines. We therefore clarify that,

(1) Complexity analysis:

Let  $p$ be the number of features, $d$ the dimension of internal node and edge embeddings, $|E|$ the number of edges, and $|V|$ the number of nodes in the input graph. For the link prediction task, STAGE consists of three steps:

1. **Fully Connected STAGE-Edge-Graph Construction:** This step requires  $O(|E| p^2)$ operations because each fully connected STAGE-edge-graph has 2p nodes.
2. **Inference (GINE Layers):** We use 2 shared layers of GINE [1] for all STAGE-edge-graphs. A single layer on one fully-connected STAGE-edge-graph has complexity $O(p d + p^2 d) = O(p^2 d)$ since we have 2p d-dimensional nodes and $(2p)^2$ d-dimensional edges in each graph. Aggregating edge embeddings across all graphs takes $O(|E| p^2 d)$.
3. **Inference (NBFNet):** We use NBFNet to perform message passing on the original graph, which requires $O(|E|d + |V|d^2)$ for one forward pass [2].

**Total Complexity:** The overall forward pass has a complexity of $O(|E| p^2 d + |E|d + |V|d^2)$.

(2) STAGE does not pose a heavy computation overhead:

To assess the computational overhead of STAGE, we measured the average wall time per training epoch on the E-Commerce Stores dataset. This dataset, with 17,463 nodes, 183,618 edges, and up to 16 node attributes, represents the largest in our experiments and thus presents the most demanding scenario for STAGE-edge-graph construction. Utilizing an 80GB A100 GPU, we obtain the following results:

| Models          	| Wall Time per Training Epoch on E-Commerce (seconds) | Zero-shot Hits@1 Performance on H&M |
|---------------------|----------------------------------------|----------------------------------------|
| NBFNet-raw      	| 318.65 | 0.0005 $\pm$ 0.0004 |
| NBFNet-gaussian 	| 322.13 | 0.0925 $\pm$ 0.0708 |
| NBFNet-structural   | 322.31 |  0.2231 $\pm$ 0.0060 |
| NBFNet-llm      	| 316.55 | 0.2302 $\pm$ 0.0015 |
| NBFNet-normalized   | 316.87 | 0.2286 $\pm$ 0.0010 |
| **NBFNet-STAGE (Ours)** | **341.36** | **0.4666 $\pm$ 0.0020** |

NBFNet-STAGE is only 7.83% slower than the fastest baseline (NBFNet-llm), a reasonable tradeoff for its performance gains. The additional time is due to computing STAGE-edge-graph embeddings during each forward pass, while building the STAGE-edge-graphs is a one-time preprocessing step. In practice, the additional factor in the complexity has never prevented us from running in the datasets we considered.

We believe we have carefully addressed all reviewers’ feedback, providing compelling evidence for the effectiveness and generalizability of our STAGE approach. We thank the reviewer for their thoughtful comments, and we believe these revisions significantly strengthen our paper. Thank you! We hope the reviewers will find these changes satisfactory to reconsider their score. We are more than happy to address any further questions.


[1] Huang et al. "Prodigy: Enabling in-context learning over graphs." NeurIPS 2024.

[2] Liu, et al. “One for all: Towards training one graph model for all classification tasks.” ICLR 2024.

[3] Chen, et al. “LLaGA: Large language and graph assistant.” ICML, 2024.

[4] Zhao et al. "Graphany: A foundation model for node classification on any graph." ArXiv 2024.

---

### Author Response · Authors · 2024-11-21
**General Response and New Experiment Results (1/2)**

We thank all reviewers for appreciating our work and providing valuable feedback. We are glad to see that our work is recognized as addressing a “novel” and “interesting” problem with important “real-world applicability” (**U75f**, **iea1**), having a “well-articulated” and “sound theoretical support” (**U75f**, **iea1**), with “extensive” and “SOTA empirical results” (**nStQ**, **JvnB**).

In the general response, we address the reviewers' common questions and present new experimental results.

> **New baselines for node classification using GraphAny**

As per reviewer JvnB's request, we clarify that:

1. Our submission included PRODIGY [1] as a baseline, presented as NBFNet-llm (link prediction, Table 1) and GINE-llm (node classification, Table 2).
2. The request for OneForAll [2] and LLaGA [3] were excluded due to incompatibility with our setup. OneForAll is limited to edge type classification (queries like (s, ?, t)) and does not support tail node prediction (queries like (s, r, ?)) or ranking metrics (e.g., Hits@1, MRR). LLaGA's reliance on LLMs is constrained by context window sizes, making it impractical for large-scale datasets like E-Commerce Store.
3. Adding an extra baseline: GraphAny [4].
Since GraphAny is specifically designed for the node classification task, we report its results in the zero-shot node classification experiment shown in Table 2 of the updated paper.

The following is the updated Table 2 incorporating GraphAny’s results:

| Models        	| Accuracy ($\uparrow$) |
|-------------------|---------------------------------|
| GINE-structural | 0.564 $\pm$ 0.0466 |
| GINE-gaussian | 0.588 $\pm$ 0.0250 |
| GINE-normalized | 0.541 $\pm$ 0.0148 |
| GINE-age | 0.582 $\pm$ 0.0657 |
| GINE-llm [3] | 0.550 $\pm$ 0.0368 |
| GraphAny [4]      	| 0.591 $\pm$ 0.0083          	|
| **GINE-STAGE (Ours)** | **0.652 $\pm$ 0.0042**          	|

We observe that GraphAny is the best-performing model among all the baseline models. Nevertheless, our model GINE-STAGE significantly outperforms GraphAny with a 10.3% relative performance improvement. This demonstrates the effectiveness and superior performance of STAGE compared to the baseline node classification foundation model.

 > **STAGE is effective across different GNN backbones**

As per reviewer JvnB and U75f’s request, we did another ablation studying the sensitivity of STAGE using different GNN backbones. We clarify that:
1.  STAGE works well with NBFNet and GINE as the backbone model when performing on link prediction and node classification tasks.
2. Below we present an extra experiment using Graph Convolutional Network (GCN)  as the backbone model on node classification.

To further demonstrate STAGE's adaptability, we replaced the GINE backbone that was used for our results shown in Table 2 with a GCN adapted to handle multi-dimensional edge attributes. Specifically, an MLP was integrated into GCN’s message-passing to process edge attributes.

The results, shown in updated Table 6, confirm that GCN-STAGE significantly outperforms all baselines, achieving a 7.33% improvement in average zero-shot test accuracy and a much smaller standard deviation, showcasing its robustness and stability.

| Model            | Accuracy ($\uparrow$) |
|------------------|----------------------------|
| GCN-structural   | 0.547 $\pm$ 0.0658              |
| GCN-gaussian     | 0.567 $\pm$ 0.0382              |
| GCN-normalized   | 0.570 $\pm$ 0.0315              |
| GCN-llm          | 0.526 $\pm$ 0.0300              |
| **GCN-STAGE (Ours)** | **0.593** $\pm$ **0.0046**          |

These experimental results show STAGE's superior performance compared to all baseline methods, irrespective of the chosen backbone GNN architecture (GINE or GCN). This consistent improvement across different architectures underscores the versatility and broad applicability of STAGE, reinforcing its position as a robust and effective framework for graph representation learning.

---

### Meta-Review · Area_Chair_m3uU · 2024-12-26

**Metareview:**

**Summary:** This paper introduces STAGE, a method designed to enable zero-shot generalization for Graph Neural Networks (GNNs) across attributed graphs with distinct attribute domains. By focusing on statistical dependencies between attributes rather than their raw values, the method aims to generalize effectively to unseen domains. The proposed framework includes STAGE-edge-graphs to model pairwise attribute dependencies and is evaluated on link prediction and node classification tasks, showcasing significant improvements over state-of-the-art baselines.


**Decision:** The paper introduces a novel framework for zero-shot generalization of GNNs using attribute dependency modeling but fails to address critical aspects that would ensure its broader applicability. Specifically, several reviewer raised concerns that due to the algorithm's complexity of $\mathcal{O}(p^2)$, its applicability to a broader range of scenarios is significantly constrained. In fact, I noticed that the datasets used in the paper all have a relatively small number of features. However, in many real-world applications, the dimensionality of node attributes is often much higher [1, 2, 3]. Similarly, another related issue is that this method relies heavily on a clear and explicit definition or encoding of the features (e.g., I don't think it can deal with word embeddings of Abstract in a citation network), which further limits the applicability of the method. For example, in the social network datasets, 54 out of 58 features were removed as they are "difficult to encode either because they are random texts input by the user or because there is no straightforward way to turn the features into totally ordered ones". In fact, Reviewer U75f also raised a similar concern, highlighting that the method was evaluated on relatively small data sets, with no exploration of challenging or diverse datasets like biomedical or geospatial graphs.
Although the authors included a discussion of the relevant limitations in the revised version, I believe this is a more fundamental issue that needs to be addressed. Based on this, I think the paper may not yet be ready for acceptance at ICLR.

[1] Tang, Jie, et al. "ArnetMiner: extraction and mining of academic social networks." Proceedings of the 14th ACM SIGKDD international conference on Knowledge Discovery and Data Mining, 2008.

[2] Shen, Xiao, et al. "Adversarial deep network embedding for cross-network node classification." Proceedings of the AAAI conference on Artificial Intelligence, 2020.

[3] Rozemberczki, Benedek, et al. "Multi-scale attributed node embedding." Journal of Complex Networks, 2021.

**Additional Comments On Reviewer Discussion:**

This is a borderline submission, with reviewer scores of 6, 5, and 5 (Due to the uninformative review comments and the lack of participation during the rebuttal and AC-reviewer discussion, the score and feedback from Reviewer nStQ were not taken into account).

The discussion phase highlighted key concerns around scalability, experimental diversity, and clarity of motivation. While the authors provided a detailed complexity analysis and addressed some experimental gaps, these efforts were insufficient to alleviate fundamental weaknesses. The reviewers remained unconvinced about the method’s generalizability, robustness, and practical relevance in diverse real-world settings. As a result, the decision to reject was reached after careful consideration of all perspectives.

---

### Decision · Program_Chairs · 2025-01-22

Reject